# Soft-Masked Diffusion Language Models

**Michael Hersche**[1], **Samuel Moor-Smith**[1,2*], **Thomas Hofmann**[2], **Abbas Rahimi**[1]

[1]IBM Research – Zurich, [2]Department of Computer Science, ETH Zürich

## Abstract

Diffusion models have demonstrated strong potential in language modeling, offering various advantages over traditional autoregressive approaches. Their ability to generate and revise entire responses in parallel enables faster generation and built-in self-correction mechanisms. Most modern diffusion-based language models employ masked diffusion, where decoding involves iteratively processing masked tokens based on a binary decision: either retaining the mask or replacing it with the predicted token. However, this binary choice discards valuable predictive information when the mask is retained. To address this limitation, we introduce *soft-masking (SM)*, a novel method that dynamically blends the embedding of the mask token with the embeddings of the top-$k$ predicted tokens from the previous decoding step, for each retained mask. This provides the model with a more informative prior, preserving context from earlier computations and allowing partial information about masked tokens to propagate beyond a single step. We propose a training methodology that efficiently adapts masked diffusion language models to incorporate SM. We demonstrate that training a 169M parameter model from scratch with SM yields superior perplexity and MAUVE scores compared to binary masking baselines. Similarly, a pretrained model can be enhanced with SM through continued pretraining. Finally, we finetune two state-of-the-art diffusion models, Dream-7B and Dream-Coder-7B, with SM. SM consistently improves performance across multiple coding benchmarks, particularly in high-throughput settings.[1]

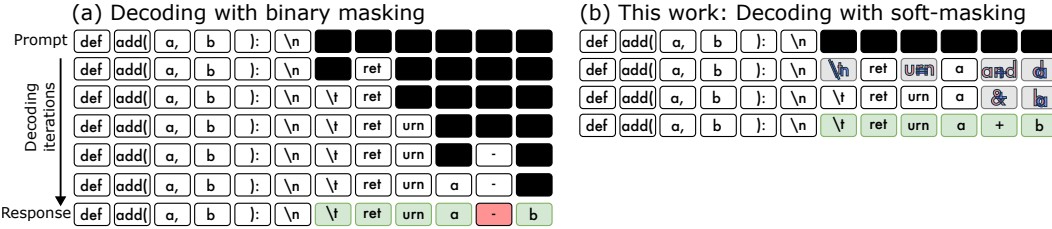

Figure 1: Illustrative answer generation using masked diffusion language models (MDLMs) via iterative decoding with (a) standard binary masking or (b) our proposed soft-masking. Our soft-masking enriches the feedback for the next decoding step by superposing the masked tokens with the previously predicted top-$k$ candidates, enabling more accurate and faster generation.

## 1 Introduction

Generative large language models (LLMs) have transformed natural language processing. LLMs typically operate in an autoregressive (AR) mode (Vaswani et al., 2017; Brown et al., 2020), where the next token in a sequence is predicted based on the previously generated tokens. While AR models have proven highly effective, their sequential nature makes inference computationally expensive,

---

*Research conducted at IBM Research – Zurich.

[1]Code at https://github.com/IBM/soft-masked-diffusion-language-models

leading to high latency and costs. These inference costs are particularly pronounced in large reasoning models (OpenAI, 2024; Guo et al., 2025; Qwen Team, 2024), where different solutions are validated through sequential exploration via chain-of-thought (CoT) (Wei et al., 2022).

As a potential remedy, recent work has shown that incorporating continuous feedback, rather than relying solely on discrete, sampled tokens, can improve AR model's performance (Hao et al., 2025; Zhuang et al., 2025; Zhang et al.). Such continuous feedback encodes multiple potential solutions in superposition (Zhu et al., 2025b), enabling simultaneous exploration of diverse paths and thereby potentially reducing the number of generated tokens. However, training AR models with continuous feedback is slow, due to the sequential reliance on previous continuous token outputs.

As a promising alternative, diffusion models—originally developed for continuous domains in vision (Sohl-Dickstein et al., 2015; Ho et al., 2020; Song et al., 2020)—have recently been adapted to natural language processing. These diffusion *language* models (DLMs) offer key advantages over AR models, including accelerated sampling (Inception et al., 2025), controllable generation (Li et al., 2022), bidirectional modeling, and built-in self-correction (Ye et al., 2024). Besides being more data-efficient than AR-models in training (Ni et al., 2025; Prabhudesai et al., 2025), they are particularly beneficial in non-causal tasks such as coding (Nie et al., 2025; Gong et al., 2025b; Xie et al., 2025). They consist of a forward process, which gradually corrupts data, and a backward process, which iteratively reverses this corruption to generate coherent outputs.

Masked DLMs (MDLMs) have emerged as the most scalable and effective approach. They implement the forward process as a categorical transition function, mapping tokens to an absorption state, typically represented by a [MASK] token (Austin et al., 2021a; Campbell et al., 2022; Lou et al., 2024; Sahoo et al., 2024; Ou et al., 2025; Shi et al., 2024). During decoding, the model makes a binary choice for each mask token: either replace it with a predicted token or retain the [MASK] (see Figure 1a). This discrete formulation allows for improved training and has enabled the development of large-scale MDLMs across both open-source (Gong et al., 2025a; Nie et al., 2025; Ye et al., 2025; Xie et al., 2025) and commercial (Inception et al., 2025; DeepMind, 2025) initiatives.

Despite their scalability, MDLMs are fundamentally constrained by the binary unmasking process, which discards valuable predictive information. Likewise, AR models commit to a single discrete sampling decision, with no opportunity for refinement. Motivated by the success of continuous feedback mechanisms in AR models, which preserve and leverage uncertainty over multiple candidates, we propose a new feedback mechanism for MDLMs that propagates this rich predictive information throughout the generation process.

**This work: Continuous feedback in MDLMs via soft-masking** We introduce soft-masking (SM), a simple yet effective mechanism for incorporating continuous feedback into MDLMs, as illustrated in Figure 1b. During decoding, SM enriches the [MASK] state with a convex combination of the top-$k$ predicted tokens, weighted by their confidence scores. This allows the model to retain and propagate partial information across decoding steps, rather than discarding it through binary masking decisions. Our method integrates seamlessly into existing MDLM architectures, requiring minimal adaptation. Our contributions are as follows:

- We propose soft-masking (SM), a novel decoding mechanism that enhances the expressiveness of the [MASK] token in MDLMs. SM only adds three additional parameters, which can be efficiently learned together with the MDLM parameters using a parallelizable training procedure that enables MDLMs to leverage the richer feedback.
- We show that a 169M-parameter MDLM with SM trained on OpenWebText can enhance both validation perplexity and MAUVE scores in unconstrained generation. Further, SM can be integrated into a pretrained MDLM via pretraining continuation for only 100k steps.
- We demonstrate that SM generalizes to large-scale MDLMs by applying it to Dream-7B (Ye et al., 2025) and Dream-Coder-7B (Xie et al., 2025). After minimal finetuning, SM improves accuracy on HumanEval and MBPP code generation benchmarks (including plus versions), particularly in high-throughput regimes with limited decoding iterations.
- SM can be readily integrated into other MDLM efficiency enhancement techniques, such as unmasking and caching. We show that SM complements an advanced unmasking scheduler, ReMDM (Wang et al., 2025), further improving unconstrained text generation quality. Moreover, SM can leverage caching and confidence-aware blockwise decoding from Fast-dLLM (Wu et al., 2026), particularly improving generations in high-throughput settings.

## 2 BACKGROUND: MASKED DIFFUSION LANGUAGE MODELS

We begin by formulating general diffusion models (Sohl-Dickstein et al., 2015) and continue with the forward and backward diffusion processes in MDLMs (Gong et al., 2025a). We denote scalars with lower-case letters ($x$), vectors with bold lower-case letters ($\mathbf{x}$), sequences of length $T$ with a colon (e.g., $\mathbf{x}_{1:T}$), matrices with bold capital letters ($\mathbf{X}$), and the transpose with $\mathbf{X}^\top$.

Diffusion models describe a *forward* diffusion process as a Markov chain that progressively corrupts the original data: $q(\mathbf{x}_{1:T}|\mathbf{x}_0) = \prod_{t=1}^T q(\mathbf{x}_t|\mathbf{x}_{t-1})$. Here, $\mathbf{x}_0 \sim q_{\text{data}}(\mathbf{x}_0)$ is drawn from the data distribution and $q(\mathbf{x}_t|\mathbf{x}_{t-1})$ describes the transition probability at step $t$. The marginalized target distribution ($q(\mathbf{x}_T)$) should be stationary and cheap to generate (e.g., a Gaussian distribution). A *reverse* diffusion process aims to reconstruct the original data with a parameterized function $p_\theta(\mathbf{x}_{0:T}) = p_\theta(\mathbf{x}_T) \prod_{t=1}^T p_\theta(\mathbf{x}_{t-1}|\mathbf{x}_t)$.

### 2.1 MDLM MODELING

**Forward corruption**   We first focus on the corruption process for a single token; the extension to sequences is discussed later. We represent language tokens as one-hot vectors $\mathbf{x} \in \{0, 1\}^{|\mathcal{V}|}$, where $|\mathcal{V}|$ represents the cardinality of the vocabulary. The transition function in MDLMs is defined such that, at each step, the token either remains unchanged or is mapped to a designated absorption state: $[\text{MASK}] \in \mathcal{V}$. The transition can be expressed as $q(\mathbf{x}_t|\mathbf{x}_{t-1}) = \text{Cat}(\mathbf{x}_t; \mathbf{Q}_t^\top \mathbf{x}_{t-1})$, where $\text{Cat}(\cdot, \mathbf{p})$ is the categorical distribution given a probability mass vector $\mathbf{p} \in \Delta^{|\mathcal{V}|-1}$, and $[\mathbf{Q}_t]_{i,j}$ denotes the transition probability from token $i$ to token $j$ at time $t$. The marginal distribution after $s$ steps is:

$$q(\mathbf{x}_s|\mathbf{x}_0) = \text{Cat}(\mathbf{x}_s|\overline{\mathbf{Q}}_s^\top \mathbf{x}_0) = \alpha_s \mathbf{x}_0 + (1 - \alpha_s)\mathbf{m},$$

where $\mathbf{m}$ is the mask token, $\overline{\mathbf{Q}}_s = \prod_{t=1}^s \mathbf{Q}_t$, and $\alpha_s$ describes the probability of retaining the original state. The schedule is typically chosen such that $\alpha_T = 0$, ensuring that the token is absorbed into the masking state with probability 1 at the final step $T$. For example, a linear masking schedule with $\alpha_t = (1 - t/T)$ is a popular choice (Austin et al., 2021a; Gong et al., 2025a; Nie et al., 2025).

**Reverse process**   Decoding aims to reverse the corruption process by iteratively denoising the data, starting from the absorbed (masked) state at time step $T$. First, note that the forward transition probability between two time steps $0 \le s < t \le T$ is given by:

$$q(\mathbf{x}_s|\mathbf{x}_t) = \text{Cat}(\mathbf{x}_s; \overline{\mathbf{Q}}_{t|s}^\top \mathbf{x}_t),$$

where $\overline{\mathbf{Q}}_{t|s} = \overline{\mathbf{Q}}_s^{-1} \overline{\mathbf{Q}}_t$ represents the transition matrix from step $t$ back to step $s$. Assuming access to the ground-truth token $\mathbf{x}_0$, the exact posterior transition from $\mathbf{x}_t$ to $\mathbf{x}_s$ can be computed via Bayes:

$$q(\mathbf{x}_s|\mathbf{x}_t, \mathbf{x}_0) = \frac{q(\mathbf{x}_t|\mathbf{x}_s)q(\mathbf{x}_s|\mathbf{x}_0)}{q(\mathbf{x}_t|\mathbf{x}_0)} = \begin{cases} \frac{\alpha_s - \alpha_t}{1 - \alpha_t}\mathbf{x}_0 + \frac{1 - \alpha_s}{1 - \alpha_t}\mathbf{m} & \text{if } \mathbf{x}_t = \mathbf{m}, \\ \mathbf{x}_0 & \text{if } \mathbf{x}_t \neq \mathbf{m}. \end{cases}$$

Since $\mathbf{x}_0$ is unknown during inference, we approximate the posterior using a learnable function $f_\theta(\mathbf{x}_t)$ that predicts the original token from the corrupted input $\hat{q}(\mathbf{x}_s|\mathbf{x}_t, \mathbf{x}_0) = p_\theta(\mathbf{x}_s|\mathbf{x}_t, f_\theta(\mathbf{x}_t))$. Here, a learnable function[2] ($f_\theta$) approximates the ground-truth ($\mathbf{x}_0$); hence, it imitates the denoising from step $t$ to step 0. Substituting the approximation into the closed-form expression yields the parametric backward transition:

$$p_\theta(\mathbf{x}_s|\mathbf{x}_t) = \frac{\alpha_s - \alpha_t}{1 - \alpha_t}f_\theta(\mathbf{x}_t) + \frac{1 - \alpha_s}{1 - \alpha_t}\mathbf{m}. \tag{1}$$

**Reverse process in natural language processing**   The input consists of sequences of $L$ tokens: $\mathbf{x}_0^{1:L}$. Decoding begins from a fully masked sequence: $\mathbf{x}_T^{1:L} = \mathbf{m}, ..., \mathbf{m}$. At each time step $t$, the current sequence estimate is passed through a bidirectional model (e.g., a non-causal Transformer (Vaswani et al., 2017; Peebles & Xie, 2023)), yielding token-wise probability distributions:

---

[2]While original denoising models use time conditioning ($f_\theta(\mathbf{x}_t, t)$), Ou et al. (2025) present a method without time conditioning. We omit time conditioning in our theoretical formulation. However, we show experimentally that SM improves MDLMs *with* (Section 4.1) and *without* time conditioning (Section 4.2).

$\mathbf{p}_{t-1}^{1:L} = g_\theta(\mathbf{x}_t^{1:L})$, where $\mathbf{p}_{t-1}^l \in \Delta^{|\mathcal{V}|-1}$ is the predicted probability mass vector on the $|\mathcal{V}|$-dimensional simplex for token $l$. Each probability mass vector ($\mathbf{p}_{t-1}^l$) is discretized using a sampling strategy (e.g., nucleus or argmax), yielding $\tilde{\mathbf{x}}_{t-1}^{1:L}$. Describing the sampling function with $h(\cdot)$, we can write the reverse process of the entire model as the functional composition of the sampling and model forward pass: $f_\theta = h \circ g_\theta$.

**Training objective**   Given a linear schedule of $\alpha_s$, the parameters ($\theta$) are optimized by minimizing:

$$\mathcal{L}(\theta) = -\mathbb{E}_{t \sim U(0,1), \mathbf{x}_0 \sim q_{\text{data}}(\cdot), \mathbf{x}_t \sim q(\cdot|\mathbf{x}_0)} \left[ \frac{1}{t} \sum_{i=1}^{L} \mathbf{1}_{\mathbf{x}_t^i = \mathbf{m}} \log\left( (\mathbf{x}_0^i)^\top g_\theta(\mathbf{x}_0^i | \mathbf{x}_t^{1:L}) \right) \right], \qquad (2)$$

where $\mathbf{1}_{\mathbf{x}_t^i = \mathbf{m}}$ is the identity function. The loss $\mathcal{L}(\theta)$ is an upper bound on the negative log likelihood of the data distribution (Shi et al., 2024; Ou et al., 2025). $U(0, 1)$ is the uniform distribution.

## 2.2   MDLMs IN PRACTICE

**Unmasking strategies**   Equation 1 suggests that the model *randomly* decides—based on the noise schedule $\alpha_t$—whether or not to replace a masked token with the predicted value $f_\theta(\mathbf{x}_t)$. However, many MDLMs use additional unmasking heuristics that improve the generation quality. One approach is to unmask a fixed number of tokens per step, guided not only by the noise schedule but also by the model's confidence. For example, at time $t$, Dream-7B (Ye et al., 2025) selects $n \approx L/T$ tokens that have the lowest entropy values. Here, $T$ is the integer number of diffusion steps. More recent methods introduce exploratory (remasking) and accelerated (aggressive unmasking) decoding stages (Wei et al., 2025; Wang et al., 2025). Rütte et al. (2025) introduce an interpolation between masked and uniform diffusion, which introduces remasking already during the training stage.

**Conditional generation**   Conditioning the generative process on a prompt ($\mathbf{c}^{1:L_c}$) is straightforward. For decoding, the prompt is simply prefixed to the (partially) masked solution at each iteration, i.e., $\mathbf{p}_{t-1}^{1:L} | \mathbf{c}^{1:L_c} = g_\theta([\mathbf{c}^{1:L_c}, \mathbf{x}_t^{1:L}])$, where only the last $L$ tokens are updated.

## 3   SOFT-MASKED DIFFUSION LANGUAGE MODELS

As elaborated above, the iterative decoding in MDLMs makes a *binary decision*: selecting either the original mask or the token predicted by the denoising model ($f_\theta$). This binary choice results in a loss of valuable contextual information for the masked tokens. To overcome this limitation, we propose soft-masking (SM). SM augments the mask with intermediate context from the previous denoising step, thereby preserving informative cues and providing a richer input for the next denoising step.

### 3.1   SOFT-MASKING (SM)

We introduce SM, illustrated in Figure 2, which enhances the denoising process in MDLMs. As shown, the overall denoising process follows the standard framework of MDLMs. However, instead of discarding previous predictions during masking or remasking, SM gently retains information from the past predictions and incorporates them into subsequent decoding steps. This provides richer feedback to guide the next denoising step. To enable this, SM relaxes the binary constraint on the feedback tokens ($\mathbf{x}_{t-1}$) provided to the denoising model, allowing them to represent a *distribution of solutions*, i.e., $\mathbf{x}_{t-1}^l \in \Delta^{|\mathcal{V}|-1}$ instead of the one-hot $\mathbf{x}_{t-1}^l \in \{0,1\}^{|V|}$.

**General formulation**   The feedback is formally defined as:

$$\mathbf{x}_{t-1}^l = \text{sm}(\hat{\mathbf{x}}_{t-1}^l, \mathbf{p}_{t-1}^l) = \begin{cases} \left(1 - \lambda(\mathbf{p}_{t-1}^l)\right) \cdot \mathbf{m} + \lambda(\mathbf{p}_{t-1}^l) \sum_{i \in \text{top-}k(\mathbf{p}_{t-1}^l)} \pi_i \cdot \mathbf{v}_i & \text{if } \hat{\mathbf{x}}_{t-1}^l = \mathbf{m}, \\ \hat{\mathbf{x}}_{t-1}^l & \text{if } \hat{\mathbf{x}}_{t-1}^l \neq \mathbf{m}, \end{cases}$$

where $\mathbf{p}_{t-1}^l$ is the probability mass vector and $\hat{\mathbf{x}}_{t-1}^l$ is the discrete output from the denoising process. SM is applied only to masked tokens; previously predicted tokens remain unchanged. SM is implemented as a convex combination of the mask token and a weighted superposition of the top-$k$ predicted tokens. Here, $\lambda \in [0, 1)$ controls the amount of feedback, $\mathbf{v}_i \in \{0, 1\}^{|\mathcal{V}|}$ is a one-hot

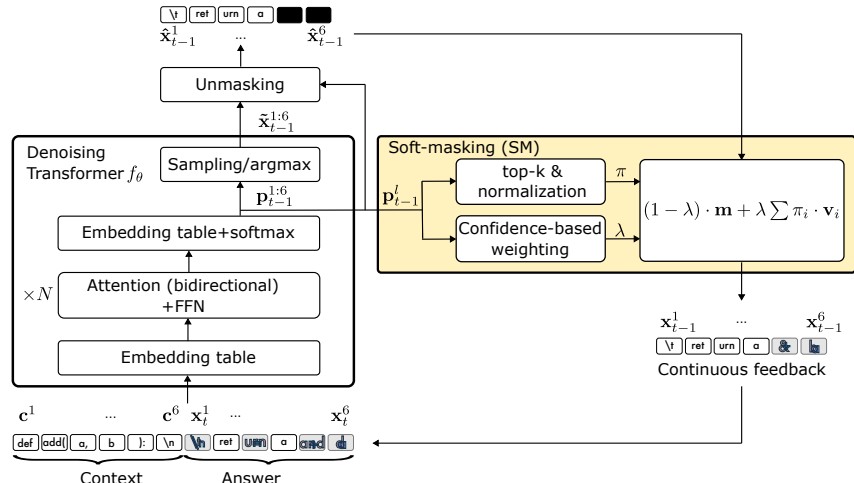

Figure 2: Iterative denoising in MDLMs using the proposed soft-masking (SM). Given a context, the aim is to predict the answer via iterative denoising of an initially fully masked response. Here, a bidirectional Transformer ($f_\theta$) performs a single denoising step. This output is passed through an unmasking function that determines which tokens remain masked. Our proposed SM enriches the masked tokens by superposing them with the normalized top-$k$ tokens at each position, weighted by a confidence parameter ($\lambda$). See Appendix A for an algorithmic description.

vector representing token $i$, and $\pi_i$ is the result of the probability mass vector being normalized over the top-$k$ tokens: $\pi_i = [\mathbf{p}^l_{t-1}]_i / \sum_{j \in \text{top-k}(\mathbf{p}^l_{t-1})} [\mathbf{p}^l_{t-1}]_j$, which ensures that $\sum_i \pi_i = 1$.

**Confidence-based weighting** In the following, we describe the dynamic weighting strategy for SM. Intuitively, a higher confidence should correspond to a greater weight on the model's output, while a lower confidence should preserve more of the original mask token. To quantify confidence, we use the negative entropy of the probability mass vector $\mathbf{p}^l_{t-1}$, denoted as $-H(\mathbf{p}^l_{t-1})$. To map this confidence score to a weight in $[0, \omega_s]$, we apply a scaled sigmoid function:

$$\lambda(\mathbf{p}_{t-1}) = \omega_s \cdot \sigma\Big(\omega_a\big(-H(\mathbf{p}^l_{t-1}) - \omega_b\big)\Big). \tag{3}$$

where $\sigma(\cdot)$ is the sigmoid function. The trainable parameters $\omega_a$, $\omega_b$, and $\omega_s$ control the steepness, the offset, and the amplitude, respectively.

## 3.2 LEARNING THE SM FEEDBACK

To teach the richer SM feedback to the model, we introduce a new training methodology that optimizes the SM parameters ($\omega$) concurrently with the main backbone model's parameters ($\theta$). This method allows the model to dynamically learn the optimal weighting between the mask token and the predicted tokens for each position, while the backbone simultaneously adapts to the richer feedback. As illustrated in Algorithm 1, the training method is a *two-pass process*. Standard MDLMs rely on the analytical tractability of the marginal distribution $q(\mathbf{x}_t|\mathbf{x}_0)$ to enable efficient single-step sampling during training. However, the SM introduces a dynamic dependency on the model's intermediate predictions, rendering the exact marginal distribution analytically intractable. Our two-pass approach serves as an approximation of this feedback-augmented marginal distribution $\tilde{q}(\mathbf{x}_t|\mathbf{x}_0)$. Specifically, we define the effective input state $\tilde{\mathbf{x}}_t = \text{sm}_\omega(\mathbf{x}_t, g_\theta(\mathbf{x}_t))$, where $\mathbf{x}_t \sim q(\cdot|\mathbf{x}_0)$, effectively maximizing the standard variational lower bound (Equation 2) using this proxy state.

First, we approximate the probability distribution of the previous denoising step by passing the corrupted data through the backbone without a gradient, yielding $\tilde{\mathbf{p}}^{1:L}_{t-1}$. This initial pass provides the necessary self-conditioning signal (the soft-masked representation) for the second pass, as practiced in (Chen et al., 2023). This distribution is then used to compute the soft-masked representation, which is passed through the backbone for a second time. The overall loss $\mathcal{L}$ from this second pass is used to update the learnable parameters for both the backbone ($\theta$) and the SM function ($\omega$) using

---

**Algorithm 1:** Training with soft-masking (SM)

---

**Input:** Backbone $g_\theta$, training corpus $q_{\text{data}}(\mathbf{x}_0)$, SM function with trainable parameters $\text{sm}_\omega$,
sampling bounds $0 \le b_l < b_h \le 1$, learning rates for backbone ($\eta_{\text{bb}}$) and SM ($\eta_{\text{sm}}$).
**Output:** Trained parameters for backbone ($\theta$) and SM parameters ($\omega$).

---

1 **repeat**
2     $\mathbf{x}_0^{1:L} \sim q_{\text{data}}(\cdot)$ ;              `// Draw samples from data distribution`
3     $t \sim \text{U}(b_l, b_h)$ ;           `// Draw time step from bounded uniform distribution`
4     $\mathbf{x}_t^{1:L} \sim q(\cdot | \mathbf{x}_0^{1:L})$ ;             `// Corrupting samples`
5     $\tilde{\theta} \leftarrow \text{detach}(\theta)$ ;          `// Generate detached copy of backbone parameters`
6     $\tilde{\mathbf{p}}_{t-1}^{1:L} \leftarrow g_{\tilde{\theta}}(\mathbf{x}_t^{1:L})$ ;          `// First model pass without gradient`
7     $\mathbf{p}_{t-1}^{1:L} \leftarrow g_\theta(\text{sm}_\omega(\mathbf{x}_t^{1:L}, \tilde{\mathbf{p}}_{t-1}^{1:L}))$ ;       `// Second model pass with SM and gradient`
8     $\mathcal{L}(\theta, \omega) \leftarrow \frac{1}{t} \sum_{l=1}^{L} \mathbf{1}_{\mathbf{x}_t^i = \mathbf{m}} \log((\mathbf{x}_0^i)^\top \mathbf{p}_{t-1}^i)$ ;       `// Compute loss`
9     Update $\theta$ and $\omega$ based on loss $\mathcal{L}$ with Adam optimizer using learning rates $\eta_{\text{bb}}$ and $\eta_{\text{sm}}$;
10 **until** *end training*;

---

their respective learning rates, $\eta_{\text{bb}}$ and $\eta_{\text{sm}}$. This approach can be highly parallelized with respect to the sequence length, unlike AR training with continuous CoT.

Arriola et al. (2024) showed that a narrower sampling interval for $t$ reduces the gradient variance when optimizing $\mathcal{L}$ with batched gradient descent. Hence, we sample from the interval $[b_l, b_h]$, $0 \le b_l < b_h \le 1$. Moreover, following the approach of Chen et al. (2023), we activate SM with probability $p_{\text{sm}} \in [0, 1]$. This prepares the model to cope with both soft-masked and standard inputs, which is particularly necessary at the beginning of the denoising process.

### 3.3 SM as an Interpolation Between Absorption and Uniform Diffusion

This section provides a conceptual interpretation of the proposed SM mechanism. To this end, we consider two extreme values of the feedback-scaling parameter ($\lambda = 0$ and $\lambda = 1$) and simplify the feedback to a single value ($k = 1$). First, assuming $\lambda = 0$ recovers vanilla MDLM. The model can always revert to this behavior by setting the scaling factor $\omega_s = 0$. Second, $\lambda = 1$ feeds the previously predicted token (based on argmax) back to the denoising model:

$$\text{sm}(\hat{\mathbf{x}}_{t-1}^l, \mathbf{p}_{t-1}^l)_{\lambda=1, k=1} = \begin{cases} \mathbf{v}_{\text{argmax}(\mathbf{p_i})} & \text{if } \hat{\mathbf{x}}_{t-1}^l = \mathbf{m}, \\ \hat{\mathbf{x}}_{t-1}^l & \text{if } \hat{\mathbf{x}}_{t-1}^l \ne \mathbf{m}. \end{cases}$$

A uniform DLM (Austin et al., 2021a) would receive the same feedback. However, the unmasking strategy remains active. Hence, this corner case can be interpreted as a masked DLM with uniform feedback for the masked states. This allows the model to explore different solutions through self-correction, enabled in the masked regions. Note SM's forward corruption process ($\lambda = 1$) deviates from the uniform formulation: SM determines the distribution $q(\mathbf{x}_t | \mathbf{x}_0)$ with the denoising model (see lines 6 and 7 in Algorithm 1) rather than from a uniform categorical sampling.

Relaxing the scaling factor to take intermediate values $\lambda \in [0, 1]$ can then be seen as an interpolation between an MDLM and a mask-augmented uniform DLM. Importantly, this interpolation occurs in the spatial embedding space. Why might it be beneficial to retain a portion of the mask token besides attenuating low-confidence predictions? One reason is that many MDLMs are pretrained to predict masked tokens, and the presence of the mask likely still carries useful positional or structural information. This effect is particularly relevant for denoising models that do not use time conditioning.

## 4 Experiments

**General setup** We begin by evaluating SM on a small-scale language modeling benchmark, using a 169M-parameter MDLM (Sahoo et al., 2024) to demonstrate its benefits with both standard and improved unmasking strategies. We then apply SM to the large-scale Dream-7B (Ye et al., 2025) and Dream-Coder-7B (Xie et al., 2025) models, showing improvements on downstream coding tasks. In addition to training from scratch, we assess the efficiency of adapting existing models via continued

Table 1: Unconstrained generation after pretraining from scratch. We report MAUVE (↑) and generative perplexity (↓) of $L = 1024$ generated tokens using MDLM (Sahoo et al., 2024) with binary masking or our SM. Evaluations are tabulated by varying NFE budgets[3]. For unmasking, we use either the standard or the more recent ReMDM (Wang et al., 2025); the highest scores are bolded. *Gain* shows the performance improvement between the SM and the baseline MDLM. [†]Results of evaluating the ground-truth data and equal-backbone AR model are taken from (Sahoo et al., 2024).

| Unmasking | Feedback | Gradient updates | Forward passes | MAUVE ↑ | | | | Generative perplexity ↓ | | | |
|---|---|---|---|---|---|---|---|---|---|---|---|
| | | | | 1/8 | 1/4 | 1/2 | 1/1 | 1/8 | 1/4 | 1/2 | 1/1 |
| Standard | Binary | 1M | 1M | 0.017 | 0.025 | 0.036 | 0.034 | 60.02 | 54.95 | 52.36 | 50.46 |
| | Our SM (iso-compute) | 0.5M | 1M | 0.143 | **0.417** | 0.498 | 0.596 | 41.08 | 31.97 | 27.36 | 24.63 |
| | *Gain* | | | *+0.126* | *+0.392* | *+0.462* | *+0.562* | *-18.93* | *-22.98* | *-24.99* | *-25.83* |
| | Our SM (iso-update) | 1M | 2M | **0.155** | 0.383 | **0.535** | **0.602** | **39.61** | **30.74** | **26.12** | **23.53** |
| | *Gain* | | | *+0.138* | *+0.358* | *+0.499* | *+0.568* | *-20.41* | *-24.21* | *-26.23* | *-26.93* |
| ReMDM | Binary | 1M | 1M | 0.075 | 0.199 | 0.292 | 0.411 | 42.53 | 31.05 | 21.75 | 28.62 |
| | Our SM (iso-compute) | 0.5M | 1M | **0.316** | **0.667** | **0.559** | 0.766 | 29.90 | 18.08 | 11.40 | 17.29 |
| | *Gain* | | | *+0.241* | *+0.468* | *+0.267* | *+0.355* | *-12.63* | *-12.97* | *-10.35* | *-11.33* |
| | Our SM (iso-update) | 1M | 2M | 0.263 | 0.626 | 0.511 | **0.774** | **29.62** | **17.58** | **10.85** | **16.72** |
| | *Gain* | | | *+0.189* | *+0.427* | *+0.219* | *+0.363* | *-12.91* | *-13.48* | *-10.90* | *-11.90* |
| AR ($T = 1024$)[†] | | 0.5M | 0.5M | 0.760 | | | | 12.1 | | | |
| Data[†] | | | | 1.0 | | | | 14.8 | | | |

Table 2: MAUVE (↑) of unconstrained generation after pretraining continuation. *Gain* shows the performance improvement between the SM and the binary MDLM with pretraining continuation.

| Unmasking | Feedback | Gradient updates | NFE budget | | | |
|---|---|---|---|---|---|---|
| | | | 1/8 | 1/4 | 1/2 | 1/1 |
| Standard | Binary | 1M | 0.017 | 0.025 | 0.036 | 0.034 |
| | Binary | 1M+100k | $0.018_{(\pm0.000)}$ | $0.027_{(\pm0.005)}$ | $0.032_{(\pm0.003)}$ | $0.038_{(\pm0.002)}$ |
| | Our SM (iso-compute) | 1M+50k | $0.054_{(\pm0.009)}$ | $0.129_{(\pm0.029)}$ | $0.200_{(\pm0.038)}$ | $\mathbf{0.259}_{(\pm0.024)}$ |
| | *Gain* | | *+0.036* | *+0.101* | *+0.168* | *+0.221* |
| | Our SM (iso-update) | 1M+100k | $\mathbf{0.059}_{(\pm0.007)}$ | $\mathbf{0.139}_{(\pm0.021)}$ | $\mathbf{0.232}_{(\pm0.026)}$ | $0.211_{(\pm0.145)}$ |
| | *Gain* | | *+0.041* | *+0.112* | *+0.200* | *+0.173* |
| ReMDM | Binary | 1M | 0.075 | 0.199 | 0.292 | 0.411 |
| | Binary | 1M+100k | $0.052_{(\pm0.005)}$ | $0.180_{(\pm0.030)}$ | $0.315_{(\pm0.032)}$ | $0.421_{(\pm0.021)}$ |
| | Our SM (iso-compute) | 1M+50k | $0.137_{(\pm0.011)}$ | $\mathbf{0.441}_{(\pm0.064)}$ | $0.610_{(\pm0.020)}$ | $\mathbf{0.693}_{(\pm0.033)}$ |
| | *Gain* | | *+0.084* | *+0.262* | *+0.295* | *+0.272* |
| | Our SM (iso-update) | 1M+100k | $\mathbf{0.146}_{(\pm0.014)}$ | $0.432_{(\pm0.035)}$ | $\mathbf{0.617}_{(\pm0.020)}$ | $0.692_{(\pm0.034)}$ |
| | *Gain* | | *+0.094* | *+0.252* | *+0.302* | *+0.271* |

pretraining (for small-scale models) or finetuning (for large-scale models). In the pretraining continuation and finetuning setup, the baseline models with binary masking are trained with the same procedure for a fair comparison. Crucially, since our proposed training algorithm (Alg. 1) requires two model forward passes per iteration (versus one in the standard training), we evaluate SM under two distinct computational budgets: (1) **Iso-update:** We match the total number of gradient updates ($N$). This isolates learning efficiency but requires roughly twice the wall-clock time for SM. (2) **Iso-compute:** We match the total number of model forward passes. In this setting, SM is trained for $N/2$ number steps, ensuring the total computational cost remains equivalent to the baseline.

## 4.1 LANGUAGE MODELING

**Setup** We evaluate SM on language modeling using a 169M-parameter MDLM (Sahoo et al., 2024) trained on the OpenWebText (OWT) (Gokaslan & Cohen, 2019). The model utilizes a Diffusion Transformer (DiT) backbone (Peebles & Xie, 2023), which integrates time-step conditioning into an encoder-only transformer. We investigate two training regimes for SM (with $k = 3$): (1) pretraining from scratch for up to 1M steps, and (2) efficient adaptation, where we apply continued pretraining to a pretrained binary MDLM for an additional 100k steps. For evaluation, we report perplexity on the OWT validation set (computed via the two-pass SM objective). As a second measure, we assess the unconstrained generation quality using both generative perplexity and the MAUVE score (Pillutla et al., 2021), the latter serving as a robust metric for diversity and quality. We vary

---

[3]The NFE budget (Appendix B.2) is the ratio between the diffusion steps and the max generation length.

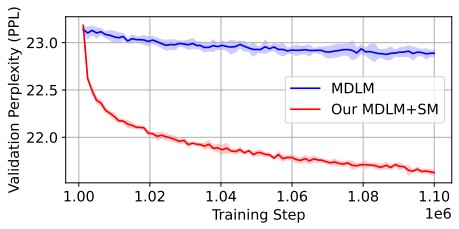

(a) Validation perplexity on OWT.

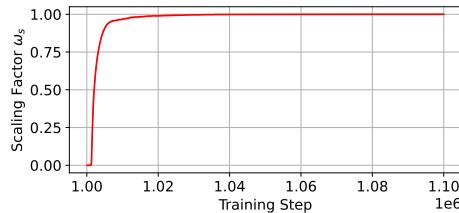

(b) SM's learned scaling factor ($\omega_s$).

Figure 3: Continuing MDLM pretraining on OpenWebText. We show the average $\pm$ standard deviation (shaded) across 5 seeds. SM is configured with $k = 3$. (a) Our SM yields better (lower) validation perplexity than binary masking. (b) The model learns to fully use SM by increasing its influence over the scaling factor $\omega_s$.

the number of function evaluations (NFE) between 128 and 1024 (representing $1/8$ to $1/1$ of the compute budget). Both the baseline MDLM and our MDLM with SM require the same number of model passes. In addition to MDLM's standard unmasking, we also evaluate the models with a recent remasking strategy (ReMDM; Wang et al. 2025). See Appendix B.1 for more details.

**Results** Table 1 shows that our SM model trained from scratch, both under the iso-compute and iso-update budgets, consistently improves the MAUVE score (by up to +0.568 points) and the generative perplexity (by up to -26.93 points) when using standard unmasking. Interestingly, the iso-compute SM model (with $N/2$=0.5M pretraining steps) even slightly outperforms the iso-update model at certain lower NFE budgets (e.g., NFE 1/4). Thus, we observe that SM is particularly effective in compute-restricted training regimes. Moreover, SM can further benefit from advanced remasking strategies, surpassing ReMDM with the binary MDLM (by up to +0.468 MAUVE points), and even outperforming the AR MAUVE score (0.760) at the highest compute budget (achieving 0.774). Table 5 in Appendix C.2 also shows that SM maintains the entropy. Additionally, our MDLM with SM achieves superior OWT validation perplexities (21.47 in iso-update and 22.36 in iso-compute), as shown in Appendix C.1.

Next, we demonstrate that SM can be efficiently integrated into a pre-existing binary MDLM via continued pretraining for up to 100k steps. As shown in Figure 3a, our SM decreases the validation perplexity on OWT (from 23.14 to 21.63). Binary MDLM (without SM) also improves the perplexity, but by a much smaller margin (from 23.14 to 22.88). Figure 3b shows that MDLM learns to make use of the richer SM feedback by increasing the scale from initially near-zero to close to 1. The gain in validation perplexity transfers to unconstrained generation (Table 2) in pretraining continuation too, where SM consistently improves the MAUVE score across all NFE budgets and unmasking strategies. Finally, all observations in MAUVE score transfer to generative perplexities and entropy, as shown in Appendix C.2.

**Ablations** We evaluate our SM design choices on language modeling in Appendix C.3. We find that using SM 80% of the time and $k = 3$ superposition yields the best validation perplexities. Moreover, an alternative SM feedback with softmax (with a trainable temperature) instead of top-k achieves competitive validation perplexities, at the cost of higher compute and memory demands. Even though the softmax allows for propagating the gradients to the first model pass, applying the update based on both model passes does not improve the perplexities while increasing the computational cost. Finally, Appendix C.5 shows that the overhead in inference by SM is small (12%).

## 4.2 CODE GENERATION

**Setup** We integrate SM into the state-of-the-art Dream-7B (Ye et al., 2025) and Dream-Coder-7B (Xie et al., 2025) instruction-tuned models. For finetuning, we aim to use the same SFT datasets as the original models. Dream-7B uses Tulu 3 (Lambert et al., 2025) and SmolLM2 (Allal et al., 2025); Dream-Coder-7B uses Ling-Coder-SFT (Codefuse & Team, 2025). We deploy parameter-efficient finetuning using weight-decomposed low-rank adaptation (DoRA; Liu et al. 2024) on nearly 270k curated training samples with a batchsize of 8, yielding 33.5k update steps. We test the models

Table 3: Accuracy (%) on coding tasks. Evaluations are tabulated by varying NFE budgets. We fine-tune the models with 5 seeds and report the mean accuracy ($\pm$ standard deviation). SM is configured with $k$=1. *Gain* shows the comparison between the SM model and the finetuned baseline. The best performing model is marked in bold. The learned SM parameters are given in Appendix C.10.

| NFE budget | Feedback | FT steps | Dream-Coder-7B (instruct) | | | | Dream-7B (instruct) | | | |
|---|---|---|---|---|---|---|---|---|---|---|
| | | | HumanEval | HumanEval+ | MBPP | MBPP+ | HumanEval | HumanEval+ | MBPP | MBPP+ |
| 1/4 | Binary | - | 25.0 | 25.0 | 27.4 | 29.4 | 18.9 | 17.1 | 26.6 | 30.2 |
| | Binary | 33.5k | 28.5$_{(\pm1.3)}$ | 27.7$_{(\pm1.8)}$ | 25.9$_{(\pm1.5)}$ | 24.6$_{(\pm1.7)}$ | 19.0$_{(\pm1.7)}$ | 15.9$_{(\pm2.8)}$ | 27.0$_{(\pm1.6)}$ | 29.2$_{(\pm1.5)}$ |
| | Our SM | 33.5k | **29.5**$_{(\pm1.8)}$ | **28.2**$_{(\pm1.7)}$ | **33.2**$_{(\pm1.8)}$ | **29.4**$_{(\pm1.9)}$ | **24.8**$_{(\pm1.8)}$ | **23.0**$_{(\pm1.3)}$ | **32.3**$_{(\pm1.3)}$ | **36.7**$_{(\pm1.0)}$ |
| | *Gain* | | *+1.0* | *+0.5* | *+7.3* | *+4.8* | *+5.8* | *+7.1* | *+5.3* | *+7.5* |
| 1/2 | Binary | - | 54.9 | 50.6 | 51.6 | 51.3 | 31.1 | 29.3 | 42.8 | 45.8 |
| | Binary | 33.5k | 53.8$_{(\pm1.4)}$ | 49.3$_{(\pm1.6)}$ | 49.8$_{(\pm0.9)}$ | 53.2$_{(\pm1.5)}$ | 33.0$_{(\pm3.0)}$ | 29.5$_{(\pm3.4)}$ | 43.1$_{(\pm0.4)}$ | 39.6$_{(\pm2.7)}$ |
| | Our SM | 33.5k | **57.2**$_{(\pm2.7)}$ | **52.6**$_{(\pm2.0)}$ | **56.2**$_{(\pm0.7)}$ | **56.4**$_{(\pm1.4)}$ | **38.3**$_{(\pm1.9)}$ | **33.8**$_{(\pm2.6)}$ | **48.4**$_{(\pm1.2)}$ | **54.7**$_{(\pm1.8)}$ |
| | *Gain* | | *+3.4* | *+3.3* | *+6.4* | *+3.2* | *+5.3* | *+4.3* | *+5.3* | *+15.1* |
| 1/1 | Binary | - | 75.0 | 69.5 | 65.8 | 70.4 | 57.9 | 53.0 | 57.8 | 63.5 |
| | Binary | 33.5k | 75.7$_{(\pm1.7)}$ | 68.9$_{(\pm2.0)}$ | 65.6$_{(\pm0.8)}$ | 68.1$_{(\pm1.1)}$ | **59.5**$_{(\pm1.8)}$ | **53.0**$_{(\pm1.0)}$ | **58.3**$_{(\pm0.1)}$ | **62.8**$_{(\pm0.7)}$ |
| | Our SM | 33.5k | **76.2**$_{(\pm1.4)}$ | **70.4**$_{(\pm1.3)}$ | **67.0**$_{(\pm0.7)}$ | **69.6**$_{(\pm0.9)}$ | 57.8$_{(\pm1.9)}$ | 50.0$_{(\pm0.7)}$ | 56.4$_{(\pm1.2)}$ | 61.9$_{(\pm0.8)}$ |
| | *Gain* | | *+0.5* | *+1.5* | *+1.4* | *+1.5* | *-1.7* | *-3.0* | *-1.9* | *-0.9* |

on two coding tasks, HumanEval (Chen et al., 2021) and MBPP (Austin et al., 2021b), as well as on their plus version (Liu et al., 2023). We report results in iso-update training, and show similar gains in iso-compute in Appendix C.7. See Appendix B.2 for more details on the experimental setup.

**Results** The mean results are given in Table 3, tabulated based on the NFE budget of the decoding. First, we see that finetuned models (without SM) can maintain the performance of the original models, validating our finetuning procedure. Second, the benefits observed in small-scale language modeling transfer to performance gains in large-scale models: the table shows a per-

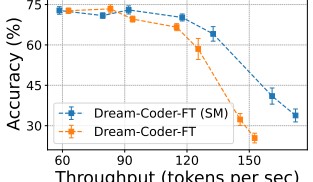

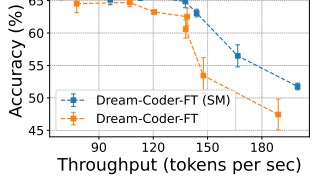

(a) HumanEval  (b) MBPP

Figure 4: Integrating SM into Fast-dLLM. We plot Dream-Coder-7B performance vs. throughput with both binary feedback and our SM. SM again excels in high-throughput settings.

formance boost on nearly all tasks (up to 15.1%). The gains are particularly prominent at lower NFE budgets. Furthermore, we show that SM complements other efficiency-enhancement mechanisms. Figure 4 reports the mean performance of the finetuned Dream-Coder-7B models, with and without SM, when combined with Fast-dLLM's blockwise caching and confidence-aware decoding (Wu et al., 2026). The performance is plotted as a function of the token throughput, which is indirectly determined from the block length of the blockwise decoding—with longer blocks correlating with higher throughputs, as described in Appendix B.2. The benefits of SM become more promising at higher throughputs.

**Ablations** To validate our design choices, we perform extensive ablation studies on the Dream models. In Appendix C.6, we evaluate SM on math tasks and notice similar performance gains. In Appendix C.8, we vary the number of top-$k$ contributions. We find that $k = 1$ is optimal for coding tasks and $k = 3$ yields the best overall results. We also tested a trainable k selection, by replacing the top-$k$ filtering with a softmax that uses a trainable temperature. However, this did not improve the performance. Finally, Appendix C.9 applies SM only in certain periods during denoising. We find that SM is particularly beneficial in the first 20% of decoding steps.

## 5 RELATED WORKS

**Continuous feedback in AR** COCONUT (Hao et al., 2025) feeds continuous token predictions back into the model to enhance reasoning capabilities. This requires training from scratch—an inherently sequential process due to its reliance on previous continuous token outputs. To reduce computational demands, many have proposed summarizing sequences of tokens into higher-level

continuous representations or concepts (LCM Team et al., 2024; Tack et al., 2026; Geiping et al., 2025). Approximate training methods such as Jacobi iterations (Wu et al., 2025) further mitigate sequential training bottlenecks by iteratively refining AR thought tokens. More recent methods feed weighted superpositions of token predictions back into the model without additional training (Zhuang et al., 2025; Zhang et al.), enabling lightweight and training-free continuous feedback. While these approaches can improve task performance, AR models often suffer from unreliable halting behavior, necessitating entropy-based heuristics to terminate decoding. In contrast, our SM introduces continuous feedback directly into MDLMs, preserving parallelism during training and inference, incurring only constant overhead during training, and avoiding halting heuristics.

**Discrete vs. continuous representations in diffusion modeling** Continuous DLMs maintain a continuous latent space throughout the diffusion process and perform a discretization step at the final readout (Li et al., 2022; Dieleman et al., 2022; Gong et al., 2022; Strudel et al., 2022; Gong et al., 2023; Gulrajani & Hashimoto, 2023); however, they generally achieve lower performance and do not allow for adaptation from pretrained AR models. Several works explore hybrid representations. HART (Tang et al., 2025) augments a discrete AR-based image predictor with a residual continuous diffusion model to correct quantization errors, but remains dependent on unidirectional AR generation, limiting self-correction. Self-conditioning (Chen et al., 2023) is a closely related approach that uses a similar two-pass training methodology in training. However, their use of concatenation increases the model complexity due to the resultant higher input dimensionality, a critical architectural difference from our method. Furthermore, it does not inherently offer a mechanism for a smooth adaptation. Sahoo et al. (2025) derive discrete uniform-state diffusion from continuous Gaussian models, enabling faster training and generation, though scalability and downstream performance remain unproven. Chao et al. (2025) propose fine-grained token representations using $l$-dimensional vectors with base-$b$ values, allowing partial unmasking during denoising, which is most effective with many decoding steps ($T \gg L$). In contrast, our SM approach improves decoding performance while maintaining constant input dimensionality. By superposing the [MASK] token and top-$k$ predictions, SM introduces continuous feedback without increasing complexity.

**Efficiency improvements of MDLMs** dLLM-Cache (Liu et al., 2025) introduces caching to MDLMs, maintaining an almost static cache for prompts and a dynamic cache for the responses. This yields a speedup of up to $9\times$ for long prompts at iso-accuracy. Semi-autoregressive generation via block diffusion (Arriola et al., 2024; Nie et al., 2025), optionally combined with caching (Wu et al., 2026), offers speedups but compromises full bidirectionality. NFE efficiency and generation quality have also been improved through dynamic unmasking strategies (Jin et al., 2025; Wei et al., 2025; Wang et al., 2025), which adapt the masking schedule during decoding. These techniques are complementary to our SM approach and can be readily integrated, as we have already demonstrated by integrating ReMDM (Wang et al., 2025) and Fast-dLLM (Wu et al., 2026) with SM.

**Concurrent works** Concurrently with the development of SM, several other works (Zheng et al., 2026; Pynadath et al., 2025; Jo et al., 2026; Zhou et al., 2025; Shariatian et al., 2025; Ma et al., 2025) sought to address information loss in MDLMs. Appendix D provides an elaborate discussion and comparison, demonstrating that SM achieves superior language modeling performance where such comparisons are feasible.

## 6 CONCLUSION

We introduced soft-masking (SM), a lightweight mechanism for incorporating continuous feedback into masked diffusion language models (MDLMs). By blending the [MASK] token with a convex combination of top-$k$ predictions during iterative decoding, SM enables more expressive and flexible updates without increasing model complexity. Applied to both small and large-scale MDLMs, SM consistently improves performance across language modeling and coding tasks. These results demonstrate that continuous feedback can enhance the capabilities of discrete diffusion models.

**Limitations and future works** Even though the training with SM is parallelizable in the sequence length, it requires an additional forward pass, which increases the complexity. We see future work in incorporating reinforcement learning-based methods (Black et al., 2023; Zhao et al., 2025; Zhu et al., 2025a) to leverage the richer feedback.

## ETHICS STATEMENT

This work does not involve human subjects, personally identifiable information, or sensitive data. All experiments were conducted using publicly available models and datasets. The proposed method is intended for research in code generation and was evaluated in controlled settings. Our focus on improving model performance in low-compute budget environments was a central consideration to mitigate environmental impact. The authors declare no known conflicts of interest.

## REPRODUCIBILITY STATEMENT

This paper describes the proposed SM method, including the training Algorithm 1 and inference Algorithm 2. The setup used for training and evaluating our model on language modeling tasks is described in Appendix B.1. The setup used for training and evaluating the Dream models for code generation is available in Appendix B.2. The code for both language modeling and code generation is provided in the supplementary materials, along with the required experimental environments.

## ACKNOWLEDGMENTS

This work was partially supported by the European Union's Horizon Europe Research and Innovation Program under Grant Agreement No. 101223271, and by the Swiss State Secretariat for Education, Research and Innovation (SERI) under contract number 25.00443. We thank Ronan Tanios for his contributions to the experimental evaluation. Moreover, we are grateful to Zaira Nazario and Abu Sebastian for the managerial support.

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

CONTENTS

---

**Algorithm 2:** Inference with soft-masking (SM)

---

**Input:** Backbone $g_\theta$, SM function with parameters $\text{sm}_\omega$, generation length $L$, number of denoising steps $T$.

**Output:** Generated sequence $\hat{\mathbf{x}}_0^{1:L}$.

---

1 $\mathbf{x}_T^{1:L} \leftarrow \mathbf{m}, \mathbf{m}, ..., \mathbf{m}$ ;            // Initialize sequence with full masks

2 **repeat**

3     $\mathbf{p}_{t-1}^{1:L} \leftarrow g_\theta\left(\mathbf{x}_t^{1:L}\right)$ ;            // Backbone pass with SM

4     $\tilde{\mathbf{x}}_{t-1}^{1:L} \leftarrow \text{sample}(\mathbf{p}_{t-1}^{1:L})$;            // Sample from backbone distribution (e.g., nucleus)

5     $\hat{\mathbf{x}}_{t-1}^{1:L} \leftarrow \text{unmask}(\tilde{\mathbf{x}}_{t-1}^{1:L}, \mathbf{p}_{t-1}^{1:L}, t, T, L)$;            // Unmasking (e.g., based on entropy)

6     $\mathbf{x}_{t-1}^{1:L} \leftarrow \text{sm}_\omega(\hat{\mathbf{x}}_t^{1:L}, \mathbf{p}_t^{1:L})$ ;            // Computing SM feedback

7     $t \leftarrow t - 1$

8 **until** *t=0*;

---

## A   Inference with SM

Algorithm 2 describes the inference procedure with SM.

## B   Experimental setup

This appendix provides details on the experiments conducted for language modeling and code generation. All experiments were run on 1–2 compute nodes with 1–8 NVIDIA A100 GPUs, each with 80 Gigabytes of VRAM.

### B.1   Language Modeling

**Pretraining from scratch**   The pretraining experiments follow the setup by Sahoo et al. (2024). We use a bidirectional Transformer backbone with 12 layers, 12 attention heads, and 768 hidden dimensions. The model is tokenized using the GPT-2 tokenizer (Radford et al., 2019). Pretraining is performed on the same OpenWebText (OWT) (Gokaslan & Cohen, 2019) split, with the last 100k documents reserved for validation. We use an AdamW optimizer with a linear learning warm-up for the first 2500 steps, and then keeping it constant at $\eta_{\text{bb}} =$3e-4 and $\eta_{\text{sm}} =$1e-2 for the backbone and the SM parameters, respectively. We train the model for 1M training steps using a batchsize of 512, which yields 262B tokens seen during training. The dropout rate is set at 0.1. Training was performed on 2 compute nodes, each with 8 A100 GPUs (80 Gigabytes of VRAM each), using a batchsize of 32 per device and deploying gradient accumulation to achieve an effective batchsize of 512.

**Pretraining continuation**   Our starting checkpoint was pretrained on OWT for 1M steps and was released by Sahoo et al. (2024). We train the model for 100k training steps using the same hyperparameters as in pretraining from scratch. We train each model with 5 different seeds (1, 2, 3, 4, 5) to account for training variability. Training was performed on 4 A100 GPUs (80 Gigabytes of VRAM each) using a batchsize of 32 per device and deploying gradient accumulation to achieve an effective batchsize of 512. Pretraining one model took approximately 64 hours and 139 hours for binary and SM, respectively.

**Soft-masking parameterization**   For the SM feedback, we add a trainable module to the model. This module contains all SM logic and augments the input embeddings with SM before the main forward process. We initialize the three SM-feedback parameters with the parameters given in Appendix B.2. We have a few imposed constraints on these values: $\omega_s \in [0, 1], \omega_a \geq 0$, and $\omega_b \leq 0$. To account for these constraints, we apply simple re-parameterizations during training:

- $\omega_s$ is passed through a sigmoid, ensuring it remains in $[0, 1]$.
- $\omega_a$ and $\omega_b$ (negative version) are each passed through a softplus, guaranteeing non-negativity.

Before inference, the learned parameters are de-parameterized: we take the direct output of the optimization, apply the inverse transforms, and for $\omega_b$ additionally negate the result so that it respects the $\omega_b \leq 0$ constraint. All other SM parameters (i.e., $k$) are specified in our added module. We simply perform a forward pass through this module to get the mixing weights for the new input embeddings.

**Generative perplexity, entropy, and MAUVE in unconstrained generation**    Our unconstrained generation evaluation follows the experimental setup by Wang et al. (2025). We perform unconstrained generation of 5000 samples ($L = 1024$) per model with a batchsize of 1 using nucleus sampling with $p = 0.9$. We use GPT-2 large for measuring the generative perplexity. Moreover, we use GPT-2 large as the embedding model for MAUVE score computation, where we set the MAUVE scaling hyperparameter to 5. Concerning ReMDM unmasking, we use the max-capped schedule ($\eta_{\text{cap}} = 0.04$) for fast sampling ($T < L$) and the loop-strategy ($t_{\text{on}} = 0.55, t_{\text{off}} = 0.05, \alpha(t_{\text{on}}) = 0.9, \eta_{\text{cap}} = 0.02$) for inference-time scaling ($T \geq L$).

## B.2    CODE GENERATION

This appendix describes the details of our finetuning experiments on Dream-7B and Dream-Coder-7B.

**Backbone models**    We use the pretrained Dream-7B (Ye et al., 2025) and Dream-Coder-7B (Xie et al., 2025) backbones, respectively. Both of these are 7B parameter models that are adapted from the Qwen2.5 family (Qwen et al., 2025; Hui et al., 2024). For both models, we use the instruction-tuned versions.

**Tasks**    We primarily evaluate on four code synthesis benchmarks:

- **HumanEval** Chen et al. (2021): a benchmark of 164 Python programming problems designed to test a model's ability to write correct and functional code from natural language specifications.

- **MBPP** Austin et al. (2021b): a benchmark consisting of 974 "Mostly Basic Programming Problems," each specified in natural language and accompanied by input–output test cases, targeting introductory-level programming tasks. We only perform evaluation with the 500 samples in the test subset. It is important to note that some works report higher scores for the Dream models on the MBPP task. This is a result of using the "sanitized" subset of the data, instead of the standard lm-eval version that we used.

- **EvalPlus: HumanEval+ and MBPP+** Liu et al. (2023): Extended benchmarks involving adding more unique test cases and correcting any inaccurate ground-truth solutions.

For our experiments, we use the standard `instruct` implementations of these benchmarks as implemented by `lm-evaluation-harness` at the time of writing. For HumanEval+, we created a custom `instruct` version of the task, using the same prompt as `humaneval_instruct`. All tasks were evaluated in a zero-shot setting with a temperature of 0.1, a top-p value of 0.9, and the entropy-based unmasking algorithm. The HumanEval tasks were evaluated with a max generation length of 768, and the MBPP tasks were evaluated with a generation length of 512.

**NFE Budget**    MDLMs typically have a maximum generation length as a parameter during generation. For our experiments, we use 768 for HumanEval(+) and 512 for MBPP(+). These models also have a parameter that quantifies the number of diffusion steps that should be taken to unveil all tokens. Typically, diffusion models set the same number of steps as the number of maximum tokens. In these cases, exactly one token is unmasked at each diffusion step.

Decreasing the number of diffusion steps leads to a much more efficient computation—by unmasking more than one token each step. From a computational cost perspective, this is essentially a linear relationship. If four tokens are unmasked at each denoising step, the generation will happen $4\times$ faster than if only one is sampled each step. This leads us to define the NFE budget of a generation:

Given a fixed-length generation task with a max number of tokens, the NFE budget of the generation will be:

$$\text{NFE budget} = \frac{\text{\# of diffusion steps}}{\text{max \# of tokens}} \tag{4}$$

In our experiments, we discuss NFE budgets of 1/4, 1/2, and 1/1. If the NFE budget is $1/n$, it means that, on average, $n$ tokens are unmasked per step.

**General setup and hyperparameters**   We aimed to keep the finetuning implementation as close to the original Dream-7B (Ye et al., 2025) SFT implementation as possible. Due to the fact that the exact SFT implementation code is unavailable, we use the DiffuLLaMa code (Gong et al., 2025a) and the Dream paper (Ye et al., 2025) for reference. Algorithm 1 displays an overview of the algorithm that we use. The only change for scaling the algorithm for finetuning larger models is that, rather than updating the full weights, we update only the weights of a light-weight parameter-efficient finetuning (PEFT) module. As our PEFT module, we use a DoRA adaptor with parameters: rank $r = 16$ and $\alpha = 16$. We apply the module only to the attention matrices: `["q_proj","k_proj","v_proj","o_proj"]`. When finetuning the SM versions, we only use the *two pass approach* and activate SM with $p_{sm} = 0.5$. We didn't see as much of an effect as language modeling with varying this value. We use an AdamW optimizer with cosine scheduling, a 0.03 warmup ratio and a max gradient norm of 7.0. The learning rates are capped at $\eta_{bb}$ =1e-5 and $\eta_{sm}$ =1e-2 for the DoRA and the SM parameters, respectively. The finetuning is performed with an effective batch size of 8 on one A100 40GB GPU and takes about 71 hours for 33.5k gradient steps.

**Training corpus**   For the training corpus, we aimed to use the same datasets that were used by the Dream models (Ye et al., 2025; Xie et al., 2025) in the SFT phase of their Instruct model training.

- **Dream-7B** The authors report instruction-tuning with *...1.8M instruction-response pairs from Tulu 3 (Lambert et al., 2025) and SmolLM2 (Allal et al., 2025)...* We use the same mix. For the Tulu 3 data, we use `allenai/tulu-3-sft-mixture`, consisting of 939,000 pairs in their *training* set. Since there is no validation set, we hold out a random 1% of these pairs for validation. For the SmolLM2 data, we use `HuggingFaceTB/smoltalk`, consisting of both a training set with 1.04M pairs and a test set with 54.9k pairs. We used the defined *test* set for validation. These datasets are the concatenated and shuffled to make up our training corpus.

- **Dream-Coder-7B** For the Coder model, Xie et al. (2025) specify explicitly that they use `inclusionAI/Ling-Coder-SFT` (Codefuse & Team, 2025) for their SFT training. We use the same dataset. This dataset consists of 4.48M pairs. We hold out a random 1% at the beginning to be used for validation.

**Preprocessing**   The preprocessing of these question-answer pairs is executed as follows:

1. **Max train/validation size.** We first sample 300,000 datapoints to use for training and 500 to use for evaluation.

2. **Context and response splitting.** All datapoints contain a sequence of *user* and *assistant* messages. We consider the last assistant message to be the *response* and all other previous messages to make up the *context*. We apply the respective models' chat template to the messages before tokenization.

3. **Filtering the dataset.** We conduct dataset filtering on three different attributes of the data: **(1)** inputs $>$ 2048 tokens; **(2)** responses $<$ 5 tokens; **(3)** any examples with tool-calling (i.e., using <tool_call> and </tool_call>). After filtering, we are typically left with just less than 270,000 context-response pairs. While this varies with our sampling seed, the total number of tokens in our training set is often just under 200 million. These 200 million tokens consist of a fifty-fifty split between prompt and response tokens.

4. **Padding with <eos> tokens.** After finetuning, we want the diffusion model to retain its ability to decide when to end its generation process. In MDLMs, this is typically done by predicting <eos> tokens for all the end positions that the model does not want to use. In order to ensure this is a part of the training process we add $n_{end} \sim \text{Uniform}(0, 50)$ <eos>

tokens to the end of each training sample. The amount of padding that is added to the training set varies for each sample.

5. **Partially masking the responses for training.** We only apply masks to the response tokens during the training process. This is performed in the following way: First, as Algorithm 1 describes, a mask probability value $t \sim \text{Uniform}(b_l, b_h)$ is sampled. This value $t$ is then used as the masking probability: a fraction $t$ of the *response* tokens are masked for the loss calculation. Arriola et al. (2024) found that sampling extremely low/high $t$ can lead to high variance in the gradient norms, making training quite difficult. For this reason, we use a clipped noise schedule of $(b_l, b_h) = (0.2, 0.8)$. The masks are sampled uniquely for each sample.

**SM-feedback parameter initialization**    As mentioned in Section 3.1, we introduced a mapping from the entropy to the amount of feedback:

$$\lambda^l(\mathbf{p}_{t-1}^l) = \omega_s \cdot \sigma\Big(\omega_a\big(-H(\mathbf{p}_{t-1}^l) - \omega_b\big)\Big). \tag{5}$$

These parameters: $\omega_a, \omega_b$ and $\omega_s$ are learnable during the training process. We ensure that the scale value, $s$, is initialized close to 0 at the start (slightly larger due to the sigmoid reparameterization discussed in Appendix B.1. This ensures that the model only adds SM-feedback if it learns that this is optimal via the finetuning process. For the steepness and center of the sigmoid, we conduct a small statistical analysis of the expected entropy distribution. Although the theoretical range of $-H(\mathbf{p}_{t-1}^l)$ is $[-\log(|\mathcal{V}|), 0]$, in practice we observed that 95% of values were above $\text{LB} \approx -1.5$. For this reason, we initialize the center of the sigmoid at $b = \text{LB}/2$ and choose $a = -10/\text{LB}$. Effectively, this normalizes the negative entropy to be between $[0, 1]$ before applying a sigmoid of $a' = 10, b' = 0.5$. We found that the learning rate for the hyperparameters needed to be set much higher than for the DoRA adaptor in order for them to be able to traverse the entire range of options. The rate was set to $\eta_{\text{sm}} = 1\text{e-}2$ for the training process.

**Fast-dLLM Throughput Experiment**    Since Fast-dLLM (Wu et al., 2026) performs blockwise decoding, there is a noticeable correlation between the block lengths and the inference time. Longer blocks allow much more parallelism in decoding and lead to a faster response, while shorter block lengths tend to drift the model towards more autoregressive generation. For our experiment, we used block lengths of 16, 32, 48, 96, 128, 192, 256, and a total generation length of 768. These led to fractional lengths of 1/48, 1/24, 1/16, 1/8, 1/6, 1/4, 1/3. We then calculated the throughput as the total tokens generated (for all questions) divided by the total decoding time. The results are shown in Figure 4.

## C    MORE RESULTS AND ABLATIONS

### C.1    LANGUAGE MODELING: OWT VALIDATION PERPLEXITY

In this appendix, we benchmark the validation perplexity of SM against state-of-the-art methods. We note that, in contrast to standard baselines, SM necessitates two model passes for perplexity evaluation. As demonstrated in Table 4, our method achieves superior performance compared to all other diffusion models in the iso-update regime, including a concurrent work (LDDM-M; Jo et al. 2026). Furthermore, even under the stricter iso-compute constraint, SM outperforms MDLM and SEDD and remains competitive with GIDD+.

Table 4: Validation perplexity on OWT. [†]Results for AR and SEDD were taken from (Sahoo et al., 2025).

|  | Gradient updates | Forward passes | Training tokens | PPL |
|---|---|---|---|---|
| AR[†] | 0.5M | 0.5M | 262B | 17.54 |
| SEDD[†] (Lou et al., 2024) | 1M | 1M | 262B | $\leq 24.10$ |
| MDLM[†] (Sahoo et al., 2024) | 1M | 1M | 262B | $\leq 23.21$ |
| GIDD+ (Rütte et al., 2025) | 1M | 1M | 262B | $\leq 22.29$ |
| LDDM-M (Jo et al., 2026) | 1M | 2M | 262B | $\leq 21.90$ |
| Our MDLM+SM (iso-compute) | 0.5M | 1M | 131B | $\leq 22.36$ |
| Our MDLM+SM (iso-update) | 1M | 2M | 262B | $\leq \mathbf{21.47}$ |

### C.2    LANGUAGE MODELING: GENERATIVE PERPLEXITY AND ENTROPY

This section analyzes the generative perplexity and the entropy observed during unconstrained generation. Table 5 shows the generative perplexity and the entropy when training MDLM with SM from scratch. The reported generative perplexity is repeated from Table 1 to facilitate comparison. As shown, our SM maintains an entropy on par with the binary masking baseline. For ReMDM unmasking, SM shows its lowest entropy at the 1/2 NFE budget (5.234 and 5.217 for iso-compute and iso-update, respectively). While still being higher than binary masking entropy (5.209), this indicates a highly low-diversity output, which likely explains the observed degradation in the MAUVE score, despite the simultaneously low generative perplexity (11.40 and 10.85). This suggests that the ReMDM unmasking process might require specific hyperparameter tuning to balance diversity with human-like text generation (MAUVE).

Table 6 and Table 7 respectively show the generative perplexity and entropy for the continuation of pretraining. Here, too, SM consistently improves the generative perplexity while maintaining the entropy.

### C.3    LANGUAGE MODELING: TRAINING SHARE, TOP-k, SOFTMAX, AND GRADIENT
### UPDATES

This appendix ablates the SM training probability ($p_{sm}$) and the top-$k$ value in the language modeling experiments. Recall that $k$ is the number of predicted tokens—generated from the previous step—that will be combined with the mask token. We start with a default configuration of $p_{sm} = 0.8$ and $k = 3$ and vary each parameter separately. As shown in Figure 5a, increasing the SM training probability from 0.5 to 0.8 improves the performance. This is likely due to the model learning to leverage continuous feedback more effectively, as a higher probability means it is exposed to the soft-masking mechanism more frequently during training. Increasing the SM training probability to 1 is detrimental, as the model loses its ability to handle binary masking, which is essential for the initial decoding steps.

Figure 5b shows that the perplexity improves when increasing $k$ from 1 to 5, with only a marginal gain observed between $k = 3$ and $k = 5$. Beyond the top-$k$ study, we investigated an alternative SM mechanism using a trainable softmax temperature. This method calculates probabilities using a learned temperature applied across the entire vocabulary, $\mathcal{V}$, letting the model scale the range of its own feedback. In Figure 5b, we denote this ablation $k = |\mathcal{V}|$ to signify that it considers the entire

Table 5: Generative perplexity (↓) and entropy (↑) with pretraining from scratch. We perform unconstrained generation of $L = 1024$ tokens using MDLM (Sahoo et al., 2024) with binary masking or our SM. Evaluations are tabulated by varying NFE budgets. For unmasking, we use either the standard or the more recent ReMDM (Wang et al., 2025); the highest scores are bolded. *Gain* shows the performance improvement between the SM and the baseline MDLM.

| Unmasking | Feedback | Gradient updates | Forward passes | Generative Perplexity ↓ | | | | Entropy ↑ | | | |
|---|---|---|---|---|---|---|---|---|---|---|---|
| | | | | 1/8 | 1/4 | 1/2 | 1/1 | 1/8 | 1/4 | 1/2 | 1/1 |
| Standard | Binary | 1M | 1M | 60.02 | 54.95 | 52.36 | 50.46 | 5.508 | 5.482 | 5.464 | 5.450 |
| | Our SM (iso-compute) | 0.5M | 1M | 41.08 | 31.97 | 27.36 | 24.63 | 5.496 | 5.448 | 5.409 | 5.374 |
| | *Gain* | | | *-18.93* | *-22.98* | *-24.99* | *-25.83* | | | | |
| | Our SM (iso-update) | 1M | 2M | **39.61** | **30.74** | **26.12** | **23.53** | 5.488 | 5.438 | 5.398 | 5.357 |
| | *Gain* | | | *-20.41* | *-24.21* | *-26.23* | *-26.93* | | | | |
| ReMDM | Binary | 1M | 1M | 42.53 | 31.05 | 21.75 | 28.62 | 5.424 | 5.336 | 5.209 | 5.368 |
| | Our SM (iso-compute) | 0.5M | 1M | 29.90 | 18.08 | 11.40 | 17.29 | 5.424 | 5.334 | 5.234 | 5.349 |
| | *Gain* | | | *-12.63* | *-12.97* | *-10.35* | *-11.33* | | | | |
| | Our SM (iso-update) | 1M | 2M | **29.62** | **17.58** | **10.85** | **16.72** | 5.416 | 5.323 | 5.217 | 5.338 |
| | *Gain* | | | *-12.91* | *-13.48* | *-10.90* | *-11.90* | | | | |
| AR ($T = 1024$) | | | | 12.1 | | | | 5.22 | | | |
| Data | | | | 14.8 | | | | 5.44 | | | |

Table 6: Generative perplexity (↓) with pretraining continuation. We perform unconstrained generation of $L = 1024$ tokens using MDLM (Sahoo et al., 2024) with binary masking or our SM. For unmasking, we use either standard or ReMDM (Wang et al., 2025); the highest scores are bolded. *Gain* shows the performance improvement between the SM and the binary MDLM with pretraining continuation.

| Unmasking | Feedback | Pretraining steps | Number of function evaluations (NFEs) | | | |
|---|---|---|---|---|---|---|
| | | | 1/8 | 1/4 | 1/2 | 1/1 |
| Standard | Binary | 1M | 60.02 | 54.95 | 52.36 | 50.46 |
| | Binary | 1M+100k | $59.99_{(\pm0.68)}$ | $54.71_{(\pm0.55)}$ | $52.15_{(\pm0.69)}$ | $50.63_{(\pm0.72)}$ |
| | Our SM (iso-compute) | 1M+50k | $51.10_{(\pm0.89)}$ | $43.25_{(\pm0.80)}$ | $39.44_{(\pm0.74)}$ | $37.46_{(\pm0.80)}$ |
| | *Gain* | | *-8.89* | *-11.46* | *-12.71* | *-13.18* |
| | Our SM (iso-update) | 1M+100k | $\mathbf{50.99}_{(\pm0.41)}$ | $\mathbf{42.75}_{(\pm0.39)}$ | $\mathbf{38.56}_{(\pm0.43)}$ | $\mathbf{35.81}_{(\pm1.24)}$ |
| | *Gain* | | *-9.00* | *-11.97* | *-13.59* | *-14.82* |
| ReMDM | Binary | 1M | 42.53 | 31.05 | 21.75 | 28.62 |
| | Binary | 1M+100k | $42.85_{(\pm0.68)}$ | $31.07_{(\pm0.39)}$ | $21.74_{(\pm0.38)}$ | $28.65_{(\pm0.33)}$ |
| | Our SM (iso-compute) | 1M+50k | $39.61_{(\pm0.87)}$ | $26.29_{(\pm0.73)}$ | $17.65_{(\pm0.47)}$ | $22.47_{(\pm0.36)}$ |
| | *Gain* | | *-3.24* | *-4.78* | *-4.08* | *-6.18* |
| | Our SM (iso-update) | 1M+100k | $\mathbf{39.52}_{(\pm0.33)}$ | $\mathbf{25.93}_{(\pm0.22)}$ | $\mathbf{17.23}_{(\pm0.16)}$ | $\mathbf{22.10}_{(\pm0.21)}$ |
| | *Gain* | | *-3.33* | *-5.14* | *-4.51* | *-6.54* |

Table 7: Entropy (↑) with pretraining continuation. We perform unconstrained generation of $L = 1024$ tokens using MDLM (Sahoo et al., 2024) with binary masking or our SM. For unmasking, we use either standard or ReMDM (Wang et al., 2025).

| Unmasking | Feedback | Pretraining steps | Number of function evaluations (NFEs) | | | |
|---|---|---|---|---|---|---|
| | | | 1/8 | 1/4 | 1/2 | 1/1 |
| Standard | Binary | 1M | 5.508 | 5.482 | 5.464 | 5.450 |
| | Binary | 1M+100k | $5.503_{(\pm0.005)}$ | $5.477_{(\pm0.006)}$ | $5.458_{(\pm0.005)}$ | $5.447_{(\pm0.007)}$ |
| | Our SM (iso-compute) | 1M+50k | $5.534_{(\pm0.004)}$ | $5.503_{(\pm0.004)}$ | $5.480_{(\pm0.003)}$ | $5.467_{(\pm0.007)}$ |
| | Our SM (iso-update) | 1M+100k | $5.542_{(\pm0.003)}$ | $5.509_{(\pm0.003)}$ | $5.485_{(\pm0.004)}$ | $5.453_{(\pm0.036)}$ |
| ReMDM | Binary | 1M | 5.424 | 5.336 | 5.209 | 5.368 |
| | Binary | 1M+100k | $5.423_{(\pm0.007)}$ | $5.333_{(\pm0.007)}$ | $5.200_{(\pm0.005)}$ | $5.361_{(\pm0.005)}$ |
| | Our SM (iso-compute) | 1M+50k | $5.476_{(\pm0.006)}$ | $5.396_{(\pm0.008)}$ | $5.302_{(\pm0.005)}$ | $5.410_{(\pm0.004)}$ |
| | Our SM (iso-update) | 1M+100k | $5.482_{(\pm0.005)}$ | $5.404_{(\pm0.004)}$ | $5.312_{(\pm0.003)}$ | $5.416_{(\pm0.005)}$ |

vocabulary. As shown, this method can slightly improve the perplexity. Crucially, the softmax mechanism is differentiable (in contrast to the non-differentiable top-$k$ selection), which allows for the backpropagation of gradients to the first model pass. However, we did not observe any further

improvements in perplexity when applying the gradients on both forward passes (see the "$k = |\mathcal{V}|$ (two updates)").

Overall, we decided to use the top-$k = 3$ option as the final configuration. The softmax approach, while theoretically interesting, comes with increased compute and memory demands (due to storing $|\mathcal{V}|$-dimensional vectors) and did not show a conclusive benefit in downstream coding performance (see Appendix C.8).

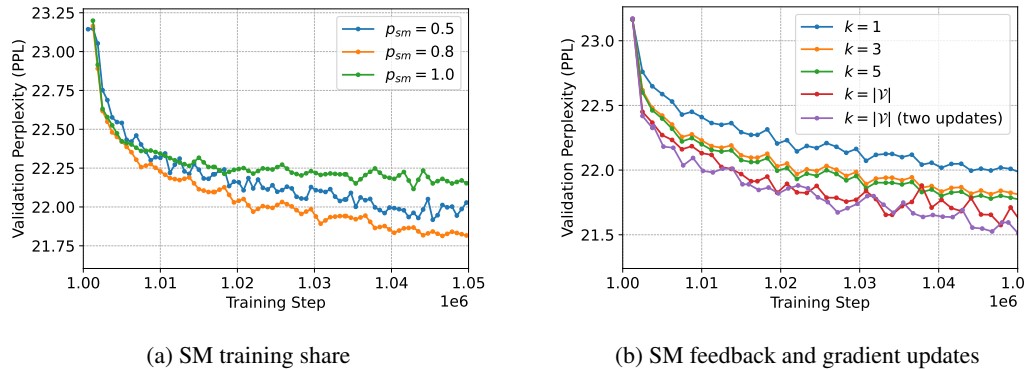

(a) SM training share                    (b) SM feedback and gradient updates

Figure 5: Ablation study language modeling on OWT. Default SM parameters are $p_{sm} = 0.8$ and $k = 3$.

## C.4 LANGUAGE MODELING: INFERENCE SPEED

This appendix analyzes the inference time for both SM and binary masking when unconditionally generating samples using $L = 1024$ diffusion steps on an NVIDIA A100 GPU. We use the inference script provided by ReMDM (Wang et al., 2025). The setup uses ancillary sampling, where the backbone's forward pass is not called if the predictive logits have not changed in the previous iteration (i.e., caching). SM is configured with $k = 3$. We measure the time using Python's cProfile.

Figure 6 shows that SM yields a small overhead of 12.5% (22.26 s vs. 19.78 s). One can see that checking the activation change ('torch.allclose') yields a major overhead in both configurations. Diving a bit deeper, one can notice that the MDLM with SM calls the backbone more often than binary masking (651 vs. 636 calls). This is likely because the more detailed input representation from SM causes the state to change more frequently, resulting in fewer cache hits. Besides, SM slightly increases the complexity on two fronts. First, computing the SM distribution requires a total of 1.61 s (0.00248 s per call), which is dominated by the top-k computation. Moreover, the sparse embedding (a weighted sum of $k + 1$ tokens) is slightly more complex than the standard embedding (a single token lookup), increasing the per-call time for the backbone by 1.7% (from 0.01728 s to 0.01757 s per call). In summary, the 2.48 s total overhead from SM is primarily composed of the 1.61 s for the SM calculation, with the remaining time due to a 1.7% increase in per-call backbone cost and a 2.4% increase in the total number of backbone calls. Despite the small overhead, the generation quality is significantly increased, as can be seen in Table 2.

## C.5 LANGUAGE MODELING: SM VISUALIZATION

This appendix provides further insight into SM's decoding dynamics. As an illustrative example, we analyze an unconstrained generation trajectory of $L = 8$ tokens over $T = 8$ steps. Note that we use a standard sampling schedule without caching or confidence-based unmasking heuristics. As shown in Figure 7, SM's confidence (indicated by color intensity) generally increases over the denoising steps. However, high SM confidence does not strictly dictate the final outcome: the top-1 predicted token within the SM evolves over time, and the final sampled token can differ from the highly confident SM prediction of the previous step (e.g., the transition from "search" to "word" at $t = 3$ for the second token).

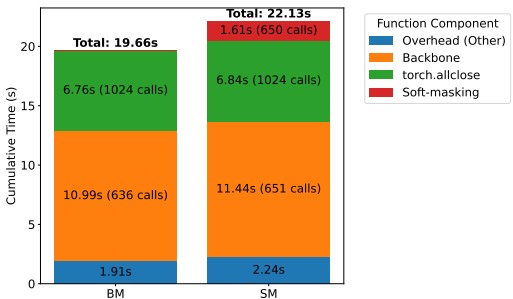

Figure 6: Cumulative time for unconditionally generating L=1024 tokens on an NVIDIA A100 GPU using standard unmasking.

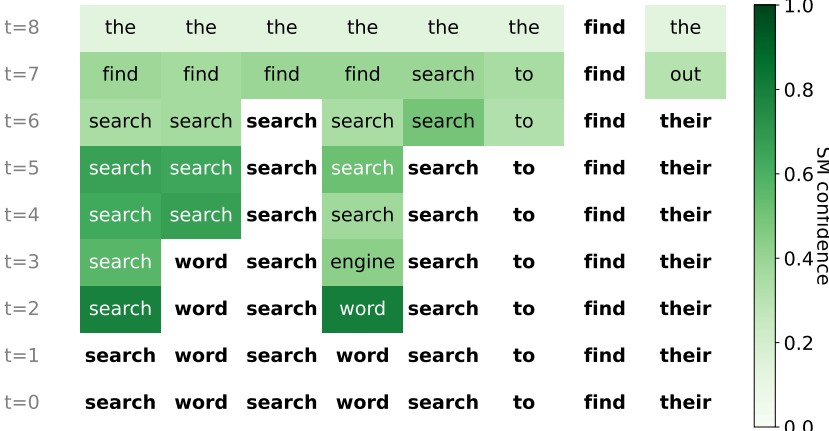

Figure 7: Unconstrained generation trajectory of $L = 8$ tokens over $T = 8$ steps using an MDLM with Soft-Masking (trained from scratch, iso-compute, 500k steps). Green-shaded cells indicate masked tokens where SM is active; color intensity corresponds to the SM confidence, with darker green indicating higher certainty. The text inside these cells displays the current top-1 predicted token. Bold, unshaded text represents tokens that have been unmasked (sampled) and fixed.

## C.6 MATHEMATICAL REASONING

We also perform experiments on mathematical reasoning tasks GSM8k (Cobbe et al., 2021) and Math-500 (Lightman et al., 2024). These evaluations are shown in Table 8. Both evaluations were performed with `lm-evaluation-harness` in a zero-shot setting. The max generation length for GSM8k was 256, and for Math-500 it was set to 512. These values were the same as the default ones used by Dream-7B in their original evaluation scripts. Since the MATH tasks are more difficult, the model often needs more generation space to come to the final answer.

## C.7 CODING: ISO-COMPUTE MODELS

Table 9 shows SM's performance in the iso-compute training setup. Note that in this more restricted setup, the model does only see half of the data, as there is no data repetition in finetuning. While the variance across seeds increased slightly, we still observe consistent gains with SM, primarily in low NFE budgets.

## C.8 CODING: TOP-$k$ AND TRAINABLE SOFTMAX TEMPERATURE

We also perform ablation tests on $k$. For these tests, we finetune four models, each with the exact same configuration, but with one key exception: we train each of these models with a different

Table 8: Accuracy (%) on math tasks. Evaluations are displayed under varying computational NFE budgets. We finetune the models with 5 seeds and report the mean accuracy (± standard deviation). SM is configured with $k$=3. *Gain* shows the comparison between the SM model and the finetuned baseline.

| NFE budget | Feedback | FT steps | GSM8k | Math-500 |
|---|---|---|---|---|
| | | | Dream-7B | |
| 1/4 | Binary | - | 57.8 | 14.2 |
| | Binary | 33.5k | $59.5_{(\pm1.1)}$ | $17.3_{(\pm1.2)}$ |
| | Our SM | 33.5k | $62.3_{(\pm2.3)}$ | $19.8_{(\pm2.1)}$ |
| | *Gain* | | *+2.8* | *+2.5* |
| 1/2 | Binary | - | 76.0 | 36.6 |
| | Binary | 33.5k | $76.5_{(\pm1.6)}$ | $36.3_{(\pm0.8)}$ |
| | Our SM | 33.5k | $79.4_{(\pm0.5)}$ | $38.8_{(\pm1.8)}$ |
| | *Gain* | | *+2.9* | *+2.5* |
| 1/1 | Binary | - | 82.0 | 45.6 |
| | Binary | 33.5k | $82.9_{(\pm1.0)}$ | $42.7_{(\pm1.4)}$ |
| | Our SM | 33.5k | $84.0_{(\pm0.7)}$ | $41.4_{(\pm1.2)}$ |
| | *Gain* | | *+1.1* | *-1.3* |

Table 9: Accuracy (%) on coding tasks. SM has been finetuned in the iso-compute training setting. Evaluations are tabulated by varying NFE budgets. We finetune the models with 5 seeds and report the mean accuracy (± standard deviation). SM is configured with $k$=1. *Gain* shows the comparison between the SM model and the finetuned baseline. The best performing model is marked in bold.

| NFE budget | Feedback | FT steps | Dream-Coder-7B (instruct) | | | | Dream-7B (instruct) | | | |
|---|---|---|---|---|---|---|---|---|---|---|
| | | | HumanEval | HumanEval+ | MBPP | MBPP+ | HumanEval | HumanEval+ | MBPP | MBPP+ |
| 1/4 | Binary | - | 25.0 | 25.0 | 27.4 | 29.4 | 18.9 | 17.1 | 26.6 | 30.2 |
| | Binary | 33.5k | $28.5_{(\pm1.3)}$ | $27.7_{(\pm1.8)}$ | $25.9_{(\pm1.5)}$ | $24.6_{(\pm1.7)}$ | $19.0_{(\pm1.7)}$ | $15.9_{(\pm2.8)}$ | $27.0_{(\pm1.6)}$ | $29.2_{(\pm1.5)}$ |
| | Our SM | 16.75k | $\mathbf{30.4}_{(\pm1.9)}$ | $\mathbf{29.0}_{(\pm1.3)}$ | $\mathbf{27.8}_{(\pm1.9)}$ | $\mathbf{28.8}_{(\pm3.7)}$ | $\mathbf{24.8}_{(\pm5.7)}$ | $\mathbf{22.4}_{(\pm4.7)}$ | $\mathbf{28.2}_{(\pm1.9)}$ | $\mathbf{36.1}_{(\pm2.0)}$ |
| | *Gain* | | *+1.9* | *+1.3* | *+1.9* | *+4.3* | *+5.8* | *+6.5* | *+1.2* | *+6.9* |
| 1/2 | Binary | - | 54.9 | 50.6 | 51.6 | 51.3 | 31.1 | 29.3 | 42.8 | 45.8 |
| | Binary | 33.5k | $53.8_{(\pm1.4)}$ | $49.3_{(\pm1.6)}$ | $\mathbf{49.8}_{(\pm0.9)}$ | $53.2_{(\pm1.5)}$ | $33.0_{(\pm3.0)}$ | $29.5_{(\pm3.4)}$ | $43.1_{(\pm0.4)}$ | $39.6_{(\pm2.7)}$ |
| | Our SM | 16.75k | $\mathbf{59.4}_{(\pm2.7)}$ | $\mathbf{54.3}_{(\pm1.9)}$ | $\mathbf{49.8}_{(\pm0.5)}$ | $\mathbf{55.2}_{(\pm1.7)}$ | $\mathbf{38.5}_{(\pm2.7)}$ | $\mathbf{33.9}_{(\pm2.8)}$ | $\mathbf{44.3}_{(\pm2.4)}$ | $\mathbf{53.6}_{(\pm2.8)}$ |
| | *Gain* | | *+5.6* | *+5.0* | *0.0* | *+2.1* | *+5.5* | *+4.4* | *+1.2* | *+14.0* |
| 1/1 | Binary | - | 75.0 | 69.5 | 65.8 | 70.4 | 57.9 | 53.0 | 57.8 | 63.5 |
| | Binary | 33.5k | $75.7_{(\pm1.7)}$ | $\mathbf{68.9}_{(\pm2.0)}$ | $65.6_{(\pm0.8)}$ | $68.1_{(\pm1.1)}$ | $\mathbf{59.5}_{(\pm1.8)}$ | $\mathbf{53.0}_{(\pm1.0)}$ | $\mathbf{58.3}_{(\pm0.1)}$ | $\mathbf{62.8}_{(\pm0.7)}$ |
| | Our SM | 16.75k | $\mathbf{75.9}_{(\pm1.4)}$ | $68.5_{(\pm0.7)}$ | $\mathbf{66.6}_{(\pm1.2)}$ | $\mathbf{68.6}_{(\pm1.2)}$ | $58.0_{(\pm2.9)}$ | $50.7_{(\pm3.3)}$ | $57.8_{(\pm1.2)}$ | $62.2_{(\pm0.8)}$ |
| | *Gain* | | *+0.2* | *-0.4* | *+1.0* | *+0.5* | *-1.5* | *-2.3* | *-0.5* | *-0.6* |

$k$-value $\in [1, 3, 5, 10]$. We use the same seed for all models to ensure the same training data and initial setup.

We also train a *fifth* model with a **trainable softmax temperature**. This method uses probabilities calculated with a learned temperature instead of using the top-k predicted tokens, letting the model scale the range of it's own feedback. In the results table, we call this ablation $k = |V|$

The results given in Table 10 illustrate a degrading performance with higher $k$ values, with $k = 1$ and $k = 3$ having the highest average performance. However, all $k$ ablations perform better than both of our baselines on average. This further illustrates the success of our proposed method.

## C.9 CODING: TIME-DEPENDENT MASKING

We explored three methods of time-dependent (TD) feedback. By time-dependence, we mean scaling the amount of SM feedback as a function of the point in the decoding process (i.e. $t$). The basic assumption here is that, the model may benefit from having more or less feedback at different steps in the diffusion process. For the following, let $t = T, ..., 1$ be our current denoising step, with $T$ being the *first* step in the *reverse* process. Let $g(\mathbf{p}) = \omega_s \cdot \sigma\left(\omega_a\left(-H(\mathbf{p}_{t-1}^l) - \omega_b\right)\right)$ be the *default* (non-time-dependent) defined in equation 3.

| Task | NFE budget | Binary feedback | | SM feedback with top-$k$ | | | | |
| | | No FT | 33.5 FT steps | 33.5k FT steps | | | | |
| | | - | - | $k=1$ | $k=3$ | $k=5$ | $k=10$ | $k=\|\mathcal{V}\|$ |
| Humaneval | 1/4 | 18.9 | 17.7 | 25.6 | 23.8 | 24.4 | 20.1 | 25.0 |
| | 1/2 | 31.1 | 33.5 | 35.4 | 36.0 | 35.4 | 34.1 | 34.1 |
| | 1/1 | 57.9 | 57.3 | 60.4 | 56.1 | 60.4 | 63.4 | 58.5 |
| MBPP | 1/4 | 26.6 | 27.2 | 32.8 | 31.4 | 27.8 | 27.6 | 27.2 |
| | 1/2 | 42.8 | 43.6 | 49.6 | 48.0 | 45.4 | 45.6 | 44.4 |
| | 1/1 | 57.8 | 58.4 | 56.2 | 57.6 | 57.2 | 56.8 | 57.8 |
| GSM8k | 1/4 | 57.8 | 60.0 | 63.3 | 62.9 | 63.3 | 60.9 | 62.3 |
| | 1/2 | 76.0 | 78.7 | 80.3 | 79.8 | 80.9 | 81.7 | 78.9 |
| | 1/1 | 82.0 | 83.5 | 82.9 | 84.6 | 84.2 | 83.6 | 84.0 |
| Math-500 | 1/4 | 14.2 | 18.6 | 21.8 | 22.0 | 20.6 | 19.2 | 20.2 |
| | 1/2 | 36.6 | 37.0 | 35.4 | 41.4 | 40.4 | 39.0 | 37.2 |
| | 1/1 | 45.6 | 42.0 | 38.4 | 41.2 | 43.2 | 43.2 | 44.4 |
| Avg. | All | 45.6 | 46.5 | 48.5 | **48.7** | 48.6 | 47.9 | 47.8 |

Table 10: Comparison of different finetuned (FT) models, each trained with a different $k$ value. The $k = |\mathcal{V}|$ ablation is trained and evaluated with a learnable softmax temperature. Evaluations are performed on all three computational budgets and four math and coding evaluation tasks. The binary baselines are also included for comparison. We see that $k = 1, 3$ and $5$ perform best, with degrading performance at higher $k$ values. The best performing model is highlighted in bold, and the second best is underlined.

1. **No TD.** In this adaptation, we apply no time-dependent feedback modification. SM is applied exactly as described above.

2. **Stepwise TD feedback function.** This time-dependent feedback defines a threshold value $0 \leq t' \leq T$. At the threshold, the model switches between SM and Binary as follows:

$$\textbf{Binary} \rightarrow \textbf{SM stepwise TD function: } \lambda^l(\mathbf{p}_{t-1}^l) = g(\mathbf{p}) \cdot \mathbf{1}_{(t \leq t')}$$

$$\textbf{SM} \rightarrow \textbf{Binary stepwise TD function: } \lambda^l(\mathbf{p}_{t-1}^l) = g(\mathbf{p}) \cdot \mathbf{1}_{(t \geq t')}$$

3. **Linear TD feedback function.** For the last formulation, we add a linear time-dependence to the feedback magnitude. Again, this entails a model switch from SM to Binary, or vice versa. The switch happens with a linear transition function:

$$\textbf{Binary} \rightarrow \textbf{SM linear TD function: } \lambda^l(\mathbf{p}_{t-1}^l) = \left(1 - \frac{t}{T}\right)[g(\mathbf{p})]$$

$$\textbf{SM} \rightarrow \textbf{Binary linear TD function: } \lambda^l(\mathbf{p}_{t-1}^l) = \left(\frac{t}{T}\right)[g(\mathbf{p})]$$

Note that these expressions are written this way because, in the reverse process, the time index $t$ *decreases* from $T \rightarrow 1$. Although we did not explicitly finetune the models to incorporate external time-dependence (TD) in the feedback function, we perform ablation studies on the TD during inference.

### C.9.1 EARLY VS. LATE STAGE SM IMPACT

We first look at a simple comparison of whether SM is more beneficial at the *early* or *late* stages of denoising. To test this, we use the **stepwise feedback function**: Specifically, we compare the **SM→Binary** and the **Binary→SM** stepwise TD functions. We define each with thresholds set such that 80% of denoising steps use SM and the remaining 20% use Binary:

1. **Binary→SM stepwise TD feedback with threshold $t' = 0.8$:** SM only during the *last* 80% of the denoising steps.

2. **SM→Binary stepwise TD feedback with threshold $t' = 0.2$:** SM only during the *first* 80% of the denoising steps.

The results given in Table 11 evaluate both approaches, comparing them with both binary baselines. We can clearly see that ablation **(2)** performs significantly better than ablation **(1)**, suggesting that SM is necessary during the early denoising steps.

| Task | Binary feedback | | SM feedback TD | |
| --- | --- | --- | --- | --- |
| | No FT | 33.5 FT steps | 33.5k FT steps | |
| | - | - | Binary→SM | SM→Binary |
| Humaneval | 18.9 | $19.0_{(\pm1.7)}$ | $17.4_{(\pm0.9)}$ | $\mathbf{24.8}_{(\pm1.1)}$ |
| MBPP | 26.6 | $27.0_{(\pm1.6)}$ | $25.2_{(\pm1.3)}$ | $\mathbf{30.9}_{(\pm0.5)}$ |
| GSM8k | 57.8 | $59.5_{(\pm1.1)}$ | $59.3_{(\pm1.1)}$ | $\mathbf{61.6}_{(\pm2.5)}$ |
| Math-500 | 14.2 | $17.3_{(\pm1.2)}$ | $18.2_{(\pm0.8)}$ | $\mathbf{20.2}_{(\pm1.9)}$ |
| Avg. | 29.4 | 31.4 | 30.0 | **34.4** |

Table 11: Comparison of whether SM is more beneficial at the early or later stages of the denoising process. Both ablations Binary→SM and SM→Binary involve a denoising process in which 80% of the steps are with SM and the remaining 20% are with binary masking. We find that placing the SM steps at the beginning of denoising results in a much greater performance boost. These ablations are compared with our fully binary baselines. The two ablation columns are the *mean* performance of the finetuned SM models with $k$=3 and a stepwise time-dependence. For each task, the best performing model is highlighted in bold. These evaluations are performed at an NFE budget of 1/4.

### C.9.2 When Is SM Most Beneficial?

After confirming that SM is necessary during the early denoising steps, we can start looking at the exact steps of the denoising where SM is more beneficial. To further understand this time-dependence, we compare five forms of our **SM→Binary** TD feedback: **(1)** no time-dependence (TD); **(2)** linear SM→Binary TD; **(3)-(5)** SM→Binary stepwise TD feedback with thresholds of $t' = 0.2$, $t' = 0.5$, and $t' = 0.8$.

The results are given in Table 12. We tabulate with the *mean* of the finetuned binary models to show that all given TD ablations of SM still perform better than the binary feedback.

| Task | Binary feedback | | SM feedback with top-$k$ | | | | |
| --- | --- | --- | --- | --- | --- | --- | --- |
| | No FT | 33.5 FT steps | 33.5k FT steps | | | | |
| | No TD | No TD | No TD | Linear | $t' = 0.2$ | $t' = 0.5$ | $t' = 0.8$ |
| Humaneval | 18.9 | $19.0_{(\pm1.7)}$ | $\mathbf{24.8}_{(\pm1.1)}$ | $22.4_{(\pm1.2)}$ | $\mathbf{24.8}_{(\pm1.1)}$ | $\mathbf{24.8}_{(\pm1.1)}$ | $\mathbf{24.8}_{(\pm1.1)}$ |
| MBPP | 26.6 | $27.0_{(\pm1.6)}$ | $30.8_{(\pm0.5)}$ | $28.4_{(\pm1.0)}$ | $\mathbf{30.9}_{(\pm0.5)}$ | $30.7_{(\pm0.5)}$ | $30.6_{(\pm0.6)}$ |
| GSM8k | 57.8 | $59.5_{(\pm1.1)}$ | $\mathbf{62.3}_{(\pm2.3)}$ | $61.6_{(\pm0.8)}$ | $61.6_{(\pm2.5)}$ | $61.8_{(\pm2.2)}$ | $59.8_{(\pm1.9)}$ |
| Math-500 | 14.2 | $17.3_{(\pm1.2)}$ | $19.8_{(\pm2.1)}$ | $\mathbf{20.4}_{(\pm1.4)}$ | $20.2_{(\pm1.9)}$ | $19.8_{(\pm1.4)}$ | $18.2_{(\pm1.8)}$ |
| Avg. | 29.4 | 31.4 | **34.4** | 33.2 | **34.4** | 34.3 | 33.3 |

Table 12: Comparison of varying forms of TD. All TD functions transition from SM→Binary with different processes. These ablations are compared with our binary baselines. All SM ablation columns are the *mean* performance of the finetuned SM models with $k$=3 and the associated TD. For each task, the best performing model is highlighted in bold.

### C.10 Coding: Learned SM-feedback Parameters

As we discussed in Section 3.1, we add the SM-feedback parameters: SM-scaling $= \omega_s$, SM-offset $= \omega_b$, and SM-steepness $= \omega_a$ to the computation graph, allowing the optimizer to train these parameters. We also add the softmax temperature for our $k = |\mathcal{V}|$ ablation (Section C.8). The

progression of these parameters throughout the full training of our SM-FT models is illustrated in Figure 8. We find that the learned scale is much lower than in the language modeling. This is likely due to the absence of time conditioning. Since the model has no other cues telling it at what stage it is at in the decoding and/or which tokens are actually masked, it relies on the existence of the mask tokens for this information.

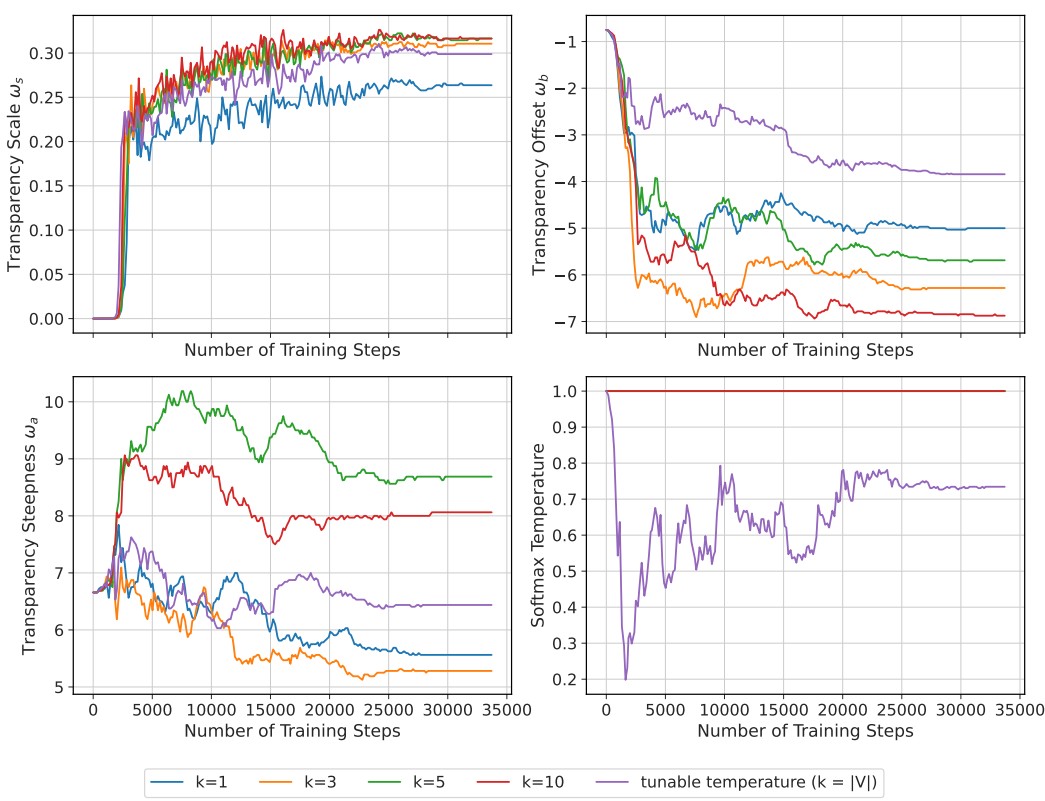

Figure 8: Plots of all four tunable hyperparameters over all of my SM-FT runs and ablations. We show the evolution of the SM-scale (top-left), the SM-offset (top-right), the SM-steepness (bottom-left), and the tunable softmax temperature (bottom-right) (for the $k = |V|$ ablation described in Section C.8).

## D  CONCURRENT WORKS

Concurrently with the development of our SM, the following works also aim to address binary MDLM's weakness in discarding valuable information during the denoising iterations.

Continuously Augmented Discrete Diffusion (CADD; Zheng et al. 2026) models the forward diffusion transition function as a product of a discrete (masking) and a continuous (Gaussian) diffusion process. This formulation permits a closed-form calculation of the forward process marginals, thereby enabling training with a single forward pass. In CADD, the input to the denoising model is a superposition of the [MASK] embedding, the embedding of the current top-1 token estimate, and a noise vector drawn from a normal distribution. The authors also propose a "soft" variant that computes a convex combination over the entire vocabulary (our $k = |\mathcal{V}|$). However, they do not show any downstream performance on language modeling and coding due to the high computational costs. In contrast, our SM covers the entire spectrum between hard (top-1) and soft through top-$k$ superposition, making a beneficial trade-off between generation quality and computational cost (see Appendix C.5). Moreover, while CADD employs a deterministic schedule to define the weighting of the estimated mean vector, our SM utilizes a learnable, confidence-based weighting mechanism. Table 13 compares the natural language modeling performance of our SM against CADD. Our SM achieves notably higher MAUVE scores in both iso-compute and iso-update settings. A similar trend

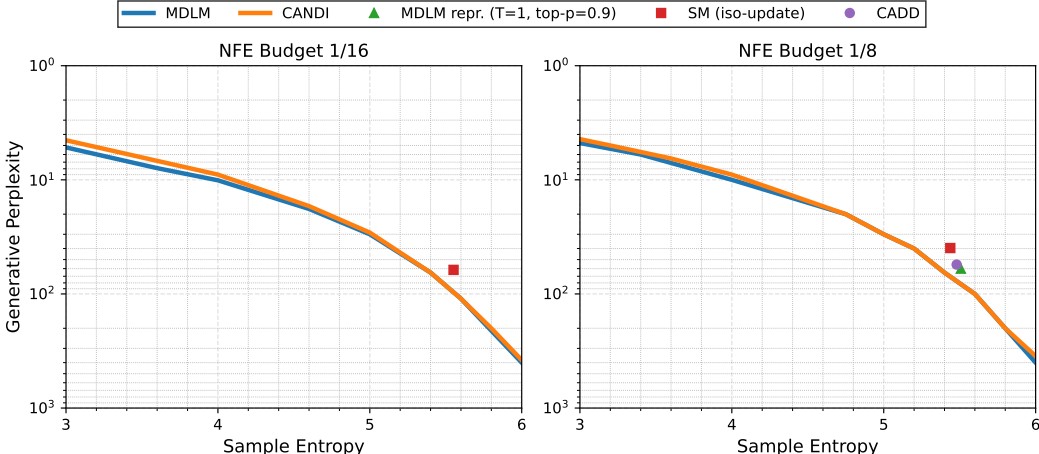

Figure 9: Comparison to CANDI (Pynadath et al., 2025) and CADD (Zheng et al., 2026) on unconstrained text generation. MDLM and CANDI results were visually extracted from (Pynadath et al., 2025). Our SM yields the best trade-off between generative perplexity and entropy.

is observed for the generative perplexity. Finally, differences in evaluation setups (qwencoder-eval vs. lm-evaluation-harness) and base models (DiffuCoder vs. DreamCoder) render a direct comparison on coding benchmarks difficult.

Table 13: Comparison to concurrent CADD (Zheng et al., 2026). All methods perform unconstrained generation with sequence length $L = 1024$. Evaluations are tabulated by varying NFE budgets. Standard unmasking is used; the highest scores are bolded.

| Feedback | Batch size | Gradient updates | Forward tokens | Loss tokens | MAUVE ↑ | | | | Generative Perplexity ↓ | | | |
|---|---|---|---|---|---|---|---|---|---|---|---|---|
| | | | | | 1/8 | 1/4 | 1/2 | 1/1 | 1/8 | 1/4 | 1/2 | 1/1 |
| MDLM | 512 | 1M | 524B | 262B | 0.017 | 0.025 | 0.036 | 0.034 | 60.0 | 55.0 | 52.4 | 50.5 |
| CADD (Zheng et al., 2026) | 256 | 2M | 524B | 262B | 0.017 | 0.080 | 0.140 | 0.240 | 55.3 | 49.6 | 46.1 | 44.6 |
| Our SM (iso-compute) | 512 | 0.5M | 524B | 131B | 0.143 | **0.417** | 0.498 | 0.596 | 41.1 | 32.0 | 27.4 | 24.6 |
| Our SM (iso-update) | 512 | 1M | 1049B | 262B | **0.155** | 0.383 | **0.535** | **0.602** | **39.6** | **30.7** | **26.1** | **23.5** |

Continuous and Discrete Diffusion (CANDI; Pynadath et al. 2025) analyzes the limitations of continuous diffusion models on discrete text through a framework based on token identifiability. Similar to CADD, CANDI merges discrete and continuous diffusion via the product of the respective transition functions. The score function is approximated via a single-step Monte-Carlo estimation, which uses the sampled token embedding. The denoising model is conditioned on a superposition of the current estimate, Gaussian noise, and a mask token, weighted by a fixed hyperparameter ($\lambda$). While allowing for closed-form marginal computation and thus single-pass training, their formulation does not involve any confidence-based weighting. Experimentally, the authors find that the unconstrained text generation is highly dependent on their sampling temperature. They show that CANDI achieves a favorable trade-off between generative perplexity and entropy when varying the temperature. Our sampling does not vary the temperature, relying solely on nucleus sampling with top-$p$=0.9, following the setup of (Wang et al., 2025). With this setup, our SM yields a superior perplexity-entropy trade-off, as shown in Figure 9.

Jo et al. (2026) describe the information loss in binary MDLMs as a sampling wall, and propose to break this wall with their Loopholing method. Loopholing directly feeds the denoising model's output latent representation (prior to readout with embedding table). The denoising model is then conditioned on the superposition of the current tokens (masked and decoded ones). Similar to SM, their training relies on self-conditioning with a conditioning probability $p$. However, Loopholing has not been successful in pretraining continuation, requiring training from scratch. We hypothesize this limitation stems from feeding back the final latents directly (causing an embedding space mismatch) without any scaling. In contrast, our learnable feedback operates in a known embedding space, enabling pretraining continuation and fine-tuning. Consequently, Loopholing results are limited to

small-scale language modeling and reasoning tasks. On OWT, they achieve a notable test perplexity improvement (21.9), which is nonetheless slightly worse than our SM (21.47) (see Table 4). Their results on unconditional text generation are not directly comparable to ours (showing overall much higher generative perplexity), which is likely due to their removal of time conditioning.

Zhou et al. (2025) theoretically show that continuous diffusion models are more expressive than masked ones, arguing that the continuous model's achievable state trajectories are a superset of those of masked model. In order to improve MDLMs, which are more popular in practice, Coevolutionary Continuous Discrete Diffusion (CCDD) proposes mixing discrete and continuous diffusion via a product of the two forward distributions. The denoising model takes as input the concatenation of the discrete and continuous representations (along the sequence dimension), predicting both the continuous and discrete representations. This doubles the input sequence length, thereby increasing compute and memory demands during training and inference. In contrast, our SM merges continuous and discrete representations via superposition with minimal computational overhead (see Appendix C.5).

Similarly, Latent Discrete Diffusion Models (LDDMs) (Shariatian et al., 2025) enrich MDLMs with latents. The discrete and latent representation can either evolve simultaneously (FUlly JoInt Denoising; FUJI) or sequentially with the latent first (SEQuential denoising; SEQ). The two channels (discrete and latent) are interconnected through a multi-modal model or additional MLPs, which increase the computational complexity.

Finally, dInfer (Ma et al., 2025) introduces iteration smoothing, which adds a convex combination of softmax probability to the masked tokens. The weighting of the feedback is increased over the decoding iterations based on a fixed schedule. While elaborate ablations are missing, authors report an effective decrease of 30-40% of decoding iterations thanks to iteration smoothing.

