# OpenReview forum: "Soft-Masked Diffusion Language Models"
_ICLR.cc/2026/Conference — ICLR 2026 Poster_

### Official Review · Reviewer_E5oT · 2025-10-31

**Soundness:** 3
**Presentation:** 3
**Contribution:** 3
**Rating:** 6
**Confidence:** 3

**Summary:**

This paper introduces a soft-masking algorithm for masked diffusion models, replacing static one-hot encodings in inputs with dynamic, context-dependent representations for mask tokens. The algorithm blends continuous feedback from an auxiliary neural network with one-hot encodings, using a confidence-based weighting strategy based on the denoising network's probability outputs at masked dimensions.

The authors assessed their methodology from two perspectives:
1. Language Modeling: They investigated if their method could improve the performance of pre-trained MDLMs through additional training steps with SM.
2. Coding Task Performance: They evaluated the method's effectiveness on coding tasks, specifically when constrained by a limited computational budget after fine-tuning.

Across both benchmarks, the proposed method demonstrated good enhancements in sample efficiency. Furthermore, the authors emphasized the consistent improvement in performance, particularly when faced with a constrained computational budget during inference.

Although the authors claim that this method is suitable for pre-training diffusion language models, no evidence about the effectiveness has been provided.

**Strengths:**

The proposed method is sound, yielding strong improvements in the fine-tuning tasks. Moreover, the evaluation is well designed.

**Weaknesses:**

While the idea and novelty are good, the analysis of the underlying mechanism of SM is insufficient. In particular, table 5 shows mixed results for SM feedback with finetuning compared to binary baselines at high compute budgets. Similarly, in Table 2, SM with finetuning underperforms Dream-7B at NFE=1/1.

My intuition is that soft-masking's "activation" of top-k tokens creates a biased learning environment, which could aid constrained inference but hinder generalization. Binary feedback, conversely, offers unbiased exploration for better generalization, albeit with higher computational cost. Analyzing soft-mask probability distribution during inference could clarify the algorithm's mechanism.

This method, if its low-compute gains stem from greedy top-k token exploitation, might be better suited for finetuning or post-training MDLMs, rather than pre-training from scratch as claimed.

Other weaknesses:

1. In Table 1, in most cases, binary feedback with additional 100k step training performs worse than the baselines with only 1M pre-training steps on MAUVE. Do the authors have any explanation for these results?

2. Following 1, the baseline results on MAUVE with binary feedback seem inconsistent with those reported in Table 1 in [1]. Please can you elaborate?

3. In addition to MAUVE, it’s worth reporting validation perplexity as well, which is a standard metric of language modeling. If the perplexity was biased by the confidence-based weighting, which is a greedy strategy, it’ll be useful to know how much bias this strategy can bring in.

4. Lacking a comprehensive comparison against a wider range of models in language modeling, including AR, other MDLMs with different model sizes.


[1] https://arxiv.org/abs/2503.00307

**Questions:**

1. Please can you also provide more detail about inference with SM?

2. What do the two sub-columns under Binary-Feedback in Table 4 and 5 correspond to?

3. To make the presentation consistent, it’ll be better to change the absolute NFE numbers in Table 1 to their corresponding ratio, same as in other Tables (3) and (4).

---

> ### Author Response · Authors · 2025-11-21
> **Response Part 1**
>
> We thank the reviewer for the positive evaluation, acknowledging novelty, and for the insightful hypothesis regarding biased learning. This motivated us to conduct a deeper analysis of the inference dynamics, which we believe significantly strengthens the paper.
>
> ---
>
> # W1: Analyzing soft-mask probability distribution during inference
> Thank you for this comment. We analyzed the token-level probability dynamics during inference (visualized in the new Appendix C.5 Figure 7). Our findings suggest that SM functions as guided exploration rather than a rigid bias.
>
> We show a generation example in the Table below, where the top-1 token in SM is represented in _italics_, and the unmasked tokens in **bold**. See Figure 7 in the revised paper for more details, including SM confidences.
>
> |     | Token 1    | Token 2  | Token 3    | Token 4  | Token 5    | Token 6 | Token 7  | Token 8   |
> |-----|------------|----------|------------|----------|------------|---------|----------|-----------|
> | t=8 |    _the_   |   _the_  |    _the_   |   _the_  |    _the_   |  _the_  | **find** |   _the_   |
> | t=7 |   _find_   |  _find_  |   _find_   |  _find_  |  _search_  |   _to_  | **find** |   _out_   |
> | t=6 |  _search_  | _search_ | **search** | _search_ |  _search_  |   _to_  | **find** | **their** |
> | t=5 |  _search_  | _search_ | **search** | _search_ | **search** |  **to** | **find** | **their** |
> | t=4 |  _search_  | _search_ | **search** | _search_ | **search** |  **to** | **find** | **their** |
> | t=3 |  _search_  | **word** | **search** | _engine_ | **search** |  **to** | **find** | **their** |
> | t=2 |  _search_  | **word** | **search** |  _word_  | **search** |  **to** | **find** | **their** |
> | t=1 | **search** | **word** | **search** | **word** | **search** |  **to** | **find** | **their** |
> | t=0 | **search** | **word** | **search** | **word** | **search** |  **to** | **find** | **their** |
>
> Our trajectory analysis reveals that SM feedback does not lock the model into a specific path. We observe frequent instances where the model overrides high-confidence SM feedback. For example, the second token is predicted as "search" for two steps, yet the model ultimately self-corrects and samples the token "word". This demonstrates that the model retains the capacity to explore and reject the enriched context provided by the SM.
>
> ---
>
> # W2: Pre-training from scratch
> Our revised version now also showcases the SM's capability to be trained from scratch. As shown in Table 1 (newly added to the paper), even at 0.5M training steps (iso-compute w.r.t. binary MDLM with 1M steps), a consistent improvement in unconditional generation is observed with respect to MAUVE scores and generative perplexity. Moreover, our MDLM with SM achieves superior OWT validation perplexities (21.47 in iso-update and 22.36 in iso-compute) in Appendix C.1. We are still waiting for the unconstrained generation results for the iso-update model (1M steps), and will update the paper as soon as the evaluation is completed.

---

> > ### Author Response · Authors · 2025-11-21
> > **Response Part 2**
> >
> > # OW1 & OW2: Baseline inconsistencies in MAUVE scores
> > Thanks for raising this. As noted in Appendix B.1, we initially used batched generation (batchsize 16) for efficiency. However, we identified an issue in the batched sampling implementation provided by Wang et al. (2025). The caching logic in the _ddpm_caching_update function is gated by a torch.allclose check across the entire batch rather than on a per-sample basis.
> >
> > This effectively disabled the intended caching mechanism. As a result, batched sampling used a functionally different algorithm, where the prediction $\mathbf{p}\_t$ was re-computed at every step, compared to sampling with a batchsize of 1, which "freezes" $\mathbf{p}\_t$ once a sample does not change.
> >
> > We switched our evaluation to eval_batch_size=1 (as in (Wang et al. 2025)), which corrects this issue, ensuring the intended sampling procedure is used. This has resolved the previously reported discrepancy in the MAUVE score, and our results now align with the baseline paper. We updated all the results (Table 1&2) with this new evaluation.
> >
> > In parallel, we will create a GitHub issue on the codebase provided by Wang et al. (2025) to raise this problem.
> >
> > ---
> >
> > # OW3: Reporting generative perplexity
> > We now also report the generative perplexity measure of the from-scratch pretrained models (new Table 1) and the pretraining continuation (Tables 5 and 6 in the new Appendix C.2). As shown, our SM also achieves notable improvement in generative perplexities, while maintaining the entropies (see Tables 4 and 7 in Appendix C.2). Hence, we cannot empirically see any bias-related negative impact of SM.
> >
> > ---
> >
> > # OW4: Comparison against other models
> > Thank you for this suggestion. We have now added the AR baseline in unconstrained generation (new Table 1) and compared the validation perplexity in the new Appendix C.1. As demonstrated in Table 4, our method achieves superior performance compared to all other diffusion models in the iso-update regime. Furthermore, even under the stricter iso-compute constraint, SM outperforms MDLM and SEDD (Lou et al. 2024) and remains competitive with GIDD+ (Rütte et al. 2025).
> >
> > ---
> >
> > # Q1: Inference with SM
> > We added an algorithmic description of the inference in a new Appendix A.
> >
> > ---
> >
> > # Q2: Headings in Tables 4 and 5
> > These correspond to no finetuning (original model) and our finetuning. We rearranged the table headings to make it clearer.
> >
> > ---
> >
> > # Q3: Consistent relative NFE
> > Agreed! We changed it to the relative NFE.

---

### Official Review · Reviewer_GihD · 2025-11-01

**Soundness:** 4
**Presentation:** 4
**Contribution:** 4
**Rating:** 8
**Confidence:** 3

**Summary:**

This worked presents soft-masked diffusion models. The idea is quite simple, during the de-noising/decoding process, have the mask embeddings at each position retain some information about the top-k possible tokens from the previous de-noising step. This method is generally applicable to most diffusion models, the authors show that you can take an off-the-shelf diffusion LM and and either continue pre-training or fine-tune with their soft-masked method to achieve performance gains with a relatively small amount of compute.

**Strengths:**

I'm very positive on this paper! The idea is very simple, but the authors conducted a fairly comprehensive amount of evaluations under various reasonable configurations and showed tangible performance gains. Overall I think this is a good paper and thus lean towards acceptance, although there are a still a few unanswered questions I would like to see answered.

**Weaknesses:**

Overall I think the paper is well executed. There are a few things that I'd like to also see to get a better idea of the limitations and benefits of this method, I'll defer those to the questions section below.

**Questions:**

1. How much cost does the does adding your SM module add to each forward pass of the model? It seems like it'd be marginal but I didn't see an exact description in this work.
2. Do the authors have any experiments on pre-training from scratch using this method? Given the results form Fig 3 (the validation PPL decreasing very fast w.r.t MDLM), it seems like MDLM+SM might lead to effeciency gains during pre-training too, do the authors have any evidence or intution about this? (Follow up, when you say "our MDLM+ SM", is the only addition the SM module or is there some other differences on MLDM loss compared to Sahoo's original work?)
3. Related to the above, how many updates steps are needed to update an MDLM model to use MDLM+SM, I see 100k steps of additional pre-training for the 169M paramedics model, how was this number chosen and generally how do the performance gains of adding +SM look for different compute budgets. This is partially touched on in the ablations, but can the authors comment a bit more on this please?
4. I'm very curious about some of the ablations that were conducted, particularly on the choice of K. Appendix B is a good start, but I'm still a bit confused on a few things. Namely (1) Why does K=1 seem to have way worse Val PPL and other K choices in Fig 5 but in Table 4 K=1,3, and 5 are very similar?  Have you noticed in your training that the choice of K is quite dependent on the task/dataset? Furthermore, in your ablations on the time dependence of SM I see that SM is most useful in the early steps (which makes intuitive sense in my opinion, SM is most useful when there are more masked tokens), but isn't the time dependence also a function of the dataset and the choice of K? I see that using a learned K wasn't particularly useful in this work, but I'd appreciate some comments on this from the perspective of a user trying to choose a reasonable K for a given task.

---

> ### Author Response · Authors · 2025-11-21
>
> We thank the reviewer for the very positive assessment and for highlighting the simplicity and effectiveness of our method. We are encouraged that you find the paper well-executed and the evaluations comprehensive.
>
> Below, we address the remaining questions to clarify the computational costs and ablation details.
>
> ---
>
> # W1: Computational overhead in forward pass
> You are correct that the overhead is marginal. We conducted an additional profiling experiment on the 168M parameter MDLM, which shows the SM module adds approximately 0.00277s (16%) per forward pass.
>
> This overhead has two components:
> -   The direct cost of computing the feedback distribution (0.00248s per call). This is dominated by the top-k operation.
> -   A minor increase in the backbone's per-call time (0.00029s) due to using a sparse embedding sum (top-k + mask) rather than a single lookup.
>
> In our L=1024 experiment, this resulted in a total inference overhead of just 12.5% (2.48s). We believe this is a modest cost for the significant quality gains. We added this detailed breakdown to Appendix C.4. Moreover, Figure 4 already shows that our SM increases the Pareto front in Throughput vs. Accuracy on coding.
>
> ---
>
> # W2: Training from scratch
> Overall, the only addition is the inclusion of our SM module; no other changes have been made to Sahoo's MDLM. Our revised version now also showcases the SM's capability to be trained from scratch. As shown in Table 1 (newly added to the paper), even at 0.5M training steps (iso-compute w.r.t. binary MDLM with 1M steps), a consistent improvement in unconditional generation is observed with respect to MAUVE scores and generative perplexity. Moreover, our MDLM with SM achieves superior OWT validation perplexities (21.47 in iso-update and 22.36 in iso-compute) in Appendix C.1. We are still waiting for the unconstrained generation results for the iso-update model (1M steps), and will update the paper as soon as the evaluation is completed.
>
> ---
>
> # W3: How much fine-tuning is needed to activate SM?
> We observe two distinct adaptation phases during pretraining continuation:
>
> 1. **SM parameter adaptation:** The SM scaling factor ($\omega_s$) initializes near zero and evolves through training. As shown in Figure 3b, this factor increases rapidly and saturates after only 20k steps for language modeling (and 25k for coding, see Figure 8).
> 2. **Backbone adaptation:** The backbone model adjusts to utilize the richer feedback. Figure 3a shows that MDLM+SM improves perplexity with a much steeper gradient than binary masking immediately after activation.
>
> To address the question on compute budgets, we added results for "half-step" training (Tables 1, 2, and 3). In all regimes (scratch, continuation, and finetuning), SM remains effective with significantly fewer updates, suggesting the training step choice in the original draft was conservative; significant gains are realized much earlier.
>
> ---
>
> # W4: On the selection of top-k.
> We do believe that there is some task dependence on optimal $k$-values. For tasks with strict constraints on individual token generation (like coding), it is likely that a lower $k$ would perform better, since there are often only a few correct options. For more flexible tasks (open language generation or freeform problem-solving tasks), having a lower $k$ may hinder exploration. Hence, as you mentioned, larger $k$ values tend to perform better in Figure 5 (for language generation tasks). As an additional experiment, we also explored the learnable top-$k$ (i.e, softmax with temperature) in language modeling (see updated Figure 5b). We found that the validation perplexity can indeed be further improved; however, this will increase memory demands and computation cost.
>
> For the time dependence, you are correct that this would also be coupled with the $k$ value, as both affect the amount of feedback provided by a predicted token. For this ablation, we intended to shed light on where the advantage of SM was most prominent. We did not notice any advantage to *not* using time dependence during the decoding process. We only found that at certain times, it was more beneficial than at others (primarily the beginning of the decoding process), which, as you mentioned, makes intuitive sense.

---

### Official Review · Reviewer_rBVW · 2025-11-02

**Soundness:** 2
**Presentation:** 3
**Contribution:** 2
**Rating:** 4
**Confidence:** 3

**Summary:**

Masked diffusion models have been recently introduced to text modeling, where the forward process is defined by transforming the valid tokens into [mask] tokens.
Despite their great potential, the mask tokens may result in information loss during the denoising process.
The authors propose a framework, called soft-masking, to assign non-trivial embeddings for the mask tokens, which can retain the semantic information of the partially decoded sequence.
The authors also give a practical training pipeline, which finetunes a pre-trained masked diffusion model.
The experiments on unconditional text generation and code generation demonstrate the superiority of SM compared to baselines.

**Strengths:**

- The presentation is clear and easy to follow.
- Although the information blank issue has been observed by concurrent works, I find the idea of using the mixture of top-k token indexes as mask token embedding is novel.
- The authors validate the effectiveness of the proposed SM method through text and code generation experiments, which supports their claim.

**Weaknesses:**

- The training methodology in Section 3.2 lacks a detailed derivation. Since the authors introduce an embedding for masked tokens, both the forward process and backward process have been changed. The authors then propose to use the two-pass method to train the the new SM model. However, this training method is rather heuristic and the authors do not provide any theoretical analysis to explain what we exactly do in the training process (e.g., maximizing the likelihood?)
- The formulation in Line 202-206 only retains the top-k probabilities on the mask position of the current prediction step. This way, it seems that SM works by memorizing the predictions that have been made previously. To exclude the memorization effect, I think an important baseline should be fixing the MDLM pre-trained model and manually tuning the parameter $\lambda$.
- As a continuation of the above point, why not use a scheme to memorize all the historical predictions, rather than only the current prediction step?
- The authors claim that the DLMs beat AR  in terms of accelerated sampling and controllable generation. Can the authors provide more evidence to support this claim?
- In line 159, what do the authors mean by $f_\theta=h\odot g_\theta$?
- As the authors do not provide a likelihood estimate formula for SM, they should clarify how they compute the perplexity score for OpenWebText.

**Questions:**

- In the training pipeline, the first model pass (Line 6, Alg 1) does not propagate the gradient. Why? How will the result change if we turn on the backpropagation in the first model pass?

---

> ### Author Response · Authors · 2025-11-21
> **Response Part 1**
>
> We thank the reviewer for the overall positive assessment of our work and for identifying critical open points, and acknowledging the novelity of the mixture of top-k token indexes as mask token embeddings. We hope that our response below successfully addresses their concerns:
>
> ---
>
> # W1. Training objective
> Thank you for raising this point. The relationship between our two-pass soft-masking (SM) training methodology and the masked diffusion objective is indeed important, and we agree that a more detailed explanation is beneficial.
>
> We confirm that the training process for the SM model still fundamentally aims to maximize the conditional likelihood $\text{log}P(\mathbf{x}_0∣\mathbf{x}_t)$ via minimizing a simplified version of the Negative Evidence Lower Bound (NELBO), as defined in the standard discrete diffusion models (Equation 2 in the paper).
>
> The primary difference between standard masked diffusion models and our SM approach is in the analytical tractability of the forward noising process.
>
> In standard masked diffusion models, the forward process is a Markov chain
> $\mathbf{x}_0 \rightarrow \mathbf{x}_1 \rightarrow ... \rightarrow \mathbf{x}_t \rightarrow ... \rightarrow \mathbf{x}_T$.
> A key property in masked diffusion models is that we can analytically compute the marginal distribution from $q(\mathbf{x}_t|\mathbf{x}_0)$. This means masking the original data with a certain probability. This is what allows for sampling a masked version $\mathbf{x}_t$ directly from $\mathbf{x}_0$ in a single step during training. This also holds for continuous diffusion models, e.g., via drawing from a normal distribution.
>
>
> Our diffusion modeling with SM does not allow for analytic computation of the marginal distribution. To this end, our two-pass method serves as an approximation of this intractable marginal distribution, which we can denote as $q' (\mathbf{x}_t | \mathbf{x}_0)$. Specifically, we construct an effective input state $\mathbf{x}'_t$ via the transformation:
>
> $$
> \mathbf{x}\_t'= sm\_\omega (g\_\theta(\mathbf{x}\_t))
> $$
>
> where $\mathbf{x}_t\sim q(\cdot| \mathbf{x}_0)$ is drawn from the standard marginal via binary masking. By optimizing the loss on the second pass using $\mathbf{x}\_t'$, we effectively minimize the NELBO using the best available approximation of the feedback-conditioned state.
>
> We have updated Section 3.2 to explicitly detail this derivation and acknowledge that this approximation strategy aligns with successful self-conditioning methods (Chen et al., 2023), which similarly utilize a preliminary pass to refine the training target.

---

> ### Author Response · Authors · 2025-11-21
> **Response Part 2**
>
> # W2. Manually tuning the feedback wheightening function
> We appreciate this insightful hypothesis. We view SM not as a mechanism for "memorization", but rather as context enrichment--providing a soft "hint" to the denoiser. To address your concern, we investigated both the training-free baseline and the decoding dynamics:
>
> - **Training-free baseline:** Actually, we initially developed a training-free method by manually tuning the feedback function. We found the optimal manual range for $\omega\_s$ to be 0.2–0.3. Remarkably, this aligns well with the values automatically learned by our models (see Appendix C.10). While this baseline showed that SM is possible without training, the results were inconsistent and did not generalize across tasks. This is likely due to a distribution shift: a standard pre-trained MDLM has never encountered soft token superpositions in its input. Fine-tuning is therefore necessary to adapt the model to effectively utilize this enriched context.
>
> - **Evidence against memorization:** In Appendix C.5, we analyzed the decoding dynamics of unconstrained generation. We show a generation example in the Table below, where the top-1 token in SM is represented in _italics_, and the unmasked tokens in **bold**. See Figure 7 in the revised paper for more details, including SM confidences. We observed that even when SM confidence is high, it does not strictly dictate the outcome of the next step. The top-1 predicted token often evolves over time (e.g., transitioning from "search" to "word" at t=3 for the second token, despite high SM confidence). This confirms that the model treats SM as a guide but retains the ability to override it when the denoising process resolves better semantic information.
>
> |     | Token 1    | Token 2  | Token 3    | Token 4  | Token 5    | Token 6 | Token 7  | Token 8   |
> |-----|------------|----------|------------|----------|------------|---------|----------|-----------|
> | t=8 |    _the_   |   _the_  |    _the_   |   _the_  |    _the_   |  _the_  | **find** |   _the_   |
> | t=7 |   _find_   |  _find_  |   _find_   |  _find_  |  _search_  |   _to_  | **find** |   _out_   |
> | t=6 |  _search_  | _search_ | **search** | _search_ |  _search_  |   _to_  | **find** | **their** |
> | t=5 |  _search_  | _search_ | **search** | _search_ | **search** |  **to** | **find** | **their** |
> | t=4 |  _search_  | _search_ | **search** | _search_ | **search** |  **to** | **find** | **their** |
> | t=3 |  _search_  | **word** | **search** | _engine_ | **search** |  **to** | **find** | **their** |
> | t=2 |  _search_  | **word** | **search** |  _word_  | **search** |  **to** | **find** | **their** |
> | t=1 | **search** | **word** | **search** | **word** | **search** |  **to** | **find** | **their** |
> | t=0 | **search** | **word** | **search** | **word** | **search** |  **to** | **find** | **their** |
>
> ---
>
> # W3. Memorizing a longer history of the trajectory
> Interesting question. It has indeed been shown that taking the entire history of states during denoising can be beneficial as a post-processing for selecting the final answer [A]. Instead, taking more samples from the past into account would break the Markovian modeling of the diffusion process and increase the memory and computation demands both in training and inference.
>
> Yet, we found the idea intriguing and developed a second-order SM formulation that takes the last two predictions into account. Formally, we defined the SM as follows:
>
> $$
> \mathbf{x}\_t = (1-\lambda) \text{sm}\_1(\hat{\mathbf{x}}\_t, \mathbf{p}\_t) + \lambda \text{sm}\_2(\hat{\mathbf{x}}\_{t+1}, \mathbf{p}\_{t+1})
> $$
>
> where $\text{sm}_1$ and $\text{sm}_2$ are SM blocks with individually learnable parameters and $\lambda$ a weighting factor. Training the model works analogously to standard SM, yet requires one more model forward pass.
>
> In our experiments, we observed no performance gain in language modeling perplexity. Notably, when λ was learnable, it consistently converged to 0 or 1, effectively collapsing the model back to a single-step dependency. We are currently running experiments with a fixed weighting factor (0.25), and will notify if there is any success.
>
> [A] Wang et al., "Time Is a Feature: Exploiting Temporal Dynamics in Diffusion Language Models," arXiv preprint arXiv:2508.09138, 2025.

---

> > ### Author Response · Authors · 2025-11-21
> > **Response Part 3**
> >
> > # W4. Where can DLMs shine over AR?
> >
> > Where to use DLMs and where to use AR models is indeed a sensible question. We believe that DLM's bidirectional modeling, its self-correction mechanism, parallel sampling, and controlled generation are the main advantages. Ergo, the tasks need to require these features. We view code generation as one of the most practical applications, but reasoning and infilling tasks can be interesting as well.
> >
> > Looking at the literature, we observe the following signs of DLM superiority over AR:
> >
> > - **Accelerated generation:** The best example of this is the commercial-grade diffusion LM, Mercury (Inception et al. 2025), which reports a 10$\times$ improvement in code generation throughput compared to frontier AR models, reaching up to 1,000 tokens per second on NVIDIA H100 GPUs.
> > - **Controllable generation:** Li et al. (2022) illustrate that DLMs perform well in various controlled generation tasks, including semantic, syntax, and length constraints. The model performs similarly (but slightly better than) an AR model trained from scratch specifically to perform infilling tasks like these.
> > - **Non-causal reasoning tasks:** Other interesting domains are planning (e.g., Sudoku) and reverse reasoning [B]. For instance, Dream-7B achieves a 4x boost over the AR Qwen2.5-7B in Sudoku tasks (Ye et al., 2025), and LlaDa-8B achieves a significant 1.25x boost over Qwen2.5-7B in a poem reversal benchmark (Nie et al., 2025).
> > - **Data-efficiency:** Furthermore, from a training standpoint, masked diffusion models are more data-efficient. This is because a single training example can yield many valid permutations of partially masked configurations, each of which can be used for training. By contrast, autoregressive models are restricted to learning only from the single left-to-right ordering of tokens (Ni, 2025) [C].
> >
> > [B] Zhang et al. "PLANNER: Generating Diversified Paragraph via Latent Language Diffusion Model," NeurIPS 2025.
> >
> > [C] Prabhudesai et al. "Diffusion Beats Autoregressive in Data-Constrained Settings," arXiv preprint arXiv:2507.15857, 2025.
> >
> > We added these references to the introduction.
> >
> > ---
> >
> > # W5. In line 159, what do the authors mean by $f_\theta= h \circ g_\theta$?
> > This is a functional composition of the backbone forward pass ($g_\theta$) and the sampling procedure ($h$). We clarified this in the updated version.
> >
> > ---
> >
> > # W6. Perplexity computation with SM
> > The validation perplexity is estimated via the ELBO. As described in Section 4.1, this requires two model passes during validation, mirroring the training process. While this increases validation compute compared to binary masking, it allows us to strictly measure the information gain provided by the SM feedback. Importantly, this lower perplexity translates directly to better generation quality in unconstrained sampling (Tables 1&2), where the inference compute cost is identical to the baseline.
> >
> > ---
> >
> > # Q1: Backpropagating through both model passes
> > Good question! As elaborated in our response to W1, the first forward model pass is used for approximating the forward diffusion process. Nevertheless, in an additional experiment, we attempted to backpropagate through both passes by replacing top-k with a differentiable softmax with a learnable temperature. As shown in updated Figure 5b, we found no significant performance benefit to justify the added computational complexity of the double-backward pass.
> >
> > Moreover, our new additional iso-compute training setup, which involves half the number of training steps, demonstrates already sufficient adaptation and strong unconditional generation and coding performance, supporting the use of single-step gradient computation and weight update.

---

### Official Review · Reviewer_wkhr · 2025-11-03

**Soundness:** 2
**Presentation:** 3
**Contribution:** 3
**Rating:** 6
**Confidence:** 4

**Summary:**

This paper introduces soft-masking, a method for improving masked diffusion LLMs by blending the mask state with a soft prediction from previous step before feeding into the denoiser network. It shows strong empirical improvements over models without soft-masking in continued pretraining.

**Strengths:**

* The motivation is strong. The work addresses a key weakness of masked diffusion LLMs - where the state is a binary decision and cannot represent superpositions of different tokens.

* The proposed fix is sensible and do not change the formulation of the probabilistic model, i.e., the model can still be trained with standard ELBO objective, with additional modification in the input of the neural network.

* The empirical improvements over pure mask models in continued pretraining is very strong (PPL 23.14 -> 21.63 vs. 23.14 -> 22.88). When applying this method to finetuning Dream coding models, The performance gain is also significant, especially in the high-throughput settings which is the main regime people prefer diffusion over AR LLMs.

**Weaknesses:**

* The proposed method requires two evaluations of the denoiser network in each training iteration. This makes the comparison to pure masked models unfair (the latter only need one network evaluation) given the same batch size. I would expect to see a comparison that matches the training flops.

* There is a very closely-related prior method "self-conditioning" that the authors failed to prominently highlight. Although it is briefly mentioned in the related work, given the smilarity of the two approaches (in fact the soft-masking can be seen as self-conditioning in discrete state spaces), in my opinion they should be discussed in more detail (the authors should point out the similarity in the method section, e.g., the two-forward-passes training).

**Questions:**

Please see above weaknesses (i will consider increasing the score if they are sufficiently addressed)

---

> ### Author Response · Authors · 2025-11-21
>
> We appreciate the reviewer's positive assessment regarding the strong motivation, sensible and novel proposed solution, and significant empirical improvements.
>
> Below, we address the raised points for improvements:
>
> ---
>
> # 1. Matching training FLOPS between binary masking and our soft-masking (SM)
> We thank the reviewer for bringing the necessity of an iso-compute comparison to our attention. While in our initial submission, we equalized the number of gradient computations between binary masking and our SM (iso-update), we now also provide results with the same number of forward passes (i.e., iso-compute). To do so, we evaluated the performance of our SM with half of the number of training steps used to train the original binary masking. Note that the training FLOPs used in our SM are now lower than in the binary masking, because: (i) we only have one gradient computation and weight-update step; (ii) SM is only applied in 80% of the cases (in 20% of the cases, we apply the standard single forward pass with binary masking).
>
> Overall, we find that SM's empirical gain remains strong even in the iso-compute training regime:
>
> 1. **Language modeling (training from scratch, 0.5M steps):**
> We now also showcase SM's capability to be trained from scratch. As shown in Table 1 (newly added to the paper), even at 0.5M training steps (iso-compute w.r.t. binary MDLM with 1M steps), a consistent improvement in unconditional generation is observed with respect to MAUVE scores and generative perplexity. The same also holds for the validation perplexity (shown in Table 4 of the new Appendix C.1). We are still waiting for the unconstrained generation results for the iso-update model (1M steps), and will update the paper again as soon as the evaluation is completed.
>
> 2. **Language modeling (continued pretraining, 50k steps):**
> We evaluated the checkpoints at 50k steps and found that they achieve MAUVE scores on par (if not even higher) in this restricted iso-compute regime. This is shown in Table 2, and the generative perplexities and entropies are reported in Appendix C.2, Tables 6 and 7.
>
> 3. **Coding (fine-tuning, 16.75k steps):**
> We restarted finetuning for 16.75k steps (iso-compute) compared to the baseline's 33.5k steps. Since this dataset is not repeated, this means our model only viewed half the training data. **Despite this reduced data coverage, the iso-compute SM model maintains performance benefits, as detailed in Table 9 (Appendix C.7).**
>
> ---
>
> # W2 Comparison to self-conditioning
> We thank the reviewer for the insightful suggestion to more prominently highlight and discuss the relationship between our SM approach and self-conditioning (Chen et al., 2023). We agree that the two approaches share the general principle of using predictive information from the previous forward pass to refine a subsequent pass.
> We are glad to note that self-conditioning inspired our choice to randomly disable SM (with 20% probability) during training (as already described in Section 3.2, Learning the SM Feedback).
>
> However, we believe that our SM significantly distinguishes itself from self-conditioning in the following ways:
>
> 1. **Discrete state spaces enriched with partially continuous feedback**: As noted by the reviewer, the two methods operate in different state spaces. Self-conditioning operates entirely in continuous state spaces, while we operate in discrete state spaces, which we augment with partially continuous feedback.
>
> 2. **Dimensionality-preserving feedback**: While self-conditioning relies on concatenation to combine the representations, which increases the model's input dimension, our SM uses **dimensionality-preserving superposition.** Our design choice efficiently reduces the computational overhead compared to direct concatenation and permits continuing pretraining or finetuning.
>
> 3. **Confidence-based weighting (learnable integration)**: We weight the previous prediction using a learnable weighting function based on the prediction's confidence. This is a key architectural feature of SM. Being initialized at near-zero, the weighting function enables a smooth integration of our SM during training (without a major validation perplexity drop even at the beginning, as shown in Figure 3a).
>
> We have made the connection to self-conditioning clearer by explicitly acknowledging the two model passes in training in Section 3.2, and by elaborating on the aforementioned differences in detail in Section 5 (Related Works).

---

### Author Response · Authors · 2025-11-21

We thank all reviewers for their constructive and insightful feedback.
We are glad that they found our proposed soft-masking method to be a sound (E5oT), novel (rBVW, E5oT), and well-motivated (wkhr) solution to the information loss problem in discrete diffusion.
Moreover, we are also encouraged that they appreciated the clarity of the paper (rBVW, GihD) and SM's strong empirical improvements (all reviewers).
The feedback has helped us to significantly improve the manuscript. In the revised version, all changes are highlighted in blue.

We would like to draw attention to three major enhancements and updates:

---

# Introducing an iso-compute training regime
While our initial submission equalized the number of gradient computations between binary masking and our SM (iso-update), reviewers wkhr and GihD motivated us to also consider training with the same number of forward passes (iso-compute). To do so, we evaluated the performance with half the number of training steps. Note that the training FLOPs used in our SM are now lower than in the binary masking, because: (i) we only have one gradient computation and weight-update step; (ii) SM is only applied in 80% of the cases (in 20% of the cases, we apply the standard single forward pass with binary masking).

Overall, we find that SM's empirical gain remains strong even in the restricted iso-compute training regime. For example, in unconstrained generation in language modeling, an iso-compute SM yields similar (if not even higher) quality samples compared to iso-update:

| Unmasking | Feedback | Updates | MAUVE 1/8 | MAUVE 1/4 | MAUVE 1/2 | MAUVE 1/1 |
| :--- | :--- | :--- | :--- | :--- | :--- | :--- |
| Standard | Binary | 1M | 0.017 | 0.025 | 0.036 | 0.034 |
| | Binary | 1M+100k | 0.018 | 0.027 | 0.032 | 0.038 |
| | Our SM (iso-compute) | 1M+50k | 0.054 | 0.129 | 0.200 | **0.259** |
| | *Gain* | | *+0.036* | *+0.101* | *+0.168* | *+0.221* |
| | Our SM (iso-update) | 1M+100k | **0.059** | **0.139** | **0.232** | 0.211 |
| | *Gain* | | *+0.041* | *+0.112* | *+0.200* | *+0.173* |
| ReMDM | Binary | 1M | 0.075 | 0.199 | 0.292 | 0.411 |
| | Binary | 1M+100k | 0.052 | 0.180 | 0.315 | 0.421 |
| | Our SM (iso-compute) | 1M+50k | 0.137 | **0.441** | 0.610 | **0.693** |
| | *Gain* | | *+0.084* | *+0.262* | *+0.295* | *+0.272* |
| | Our SM (iso-update) | 1M+100k | **0.146** | 0.432 | **0.617** | 0.692 |
| | *Gain* | | *+0.094* | *+0.252* | *+0.302* | *+0.271* |

---

# Pretraining from scratch
Based on the reviews by GihD and E5oT, our revised version now also showcases SM's capability to be trained from scratch in language modeling. Even at 0.5M training steps (iso-compute w.r.t. binary MDLM with 1M steps), a consistent improvement in unconditional generation is observed with respect to MAUVE scores and generative perplexity:

| Unmasking | Feedback | Updates | Fwd passes | MAUVE 1/8 | MAUVE 1/4 | MAUVE 1/2 | MAUVE 1/1 | PPL 1/8 | PPL 1/4 | PPL 1/2 | PPL 1/1 |
| :--- | :--- | :--- | :--- | :--- | :--- | :--- | :--- | :--- | :--- | :--- | :--- |
| Standard | Binary | 1M | 1M | 0.017 | 0.025 | 0.036 | 0.034 | 60.02 | 54.95 | 52.36 | 50.46 |
| | Our SM (iso-compute) | 0.5M | 1M | **0.143** | **0.417** | **0.498** | **0.596** | **41.08** | **31.97** | **27.36** | **24.63** |
| | *Gain* | | | *+0.126* | *+0.392* | *+0.462* | *+0.562* | *-18.93* | *-22.98* | *-24.99* | *-25.83* |
| ReMDM | Binary | 1M | 1M | 0.075 | 0.199 | 0.292 | 0.411 | 42.53 | 31.05 | 21.75 | 28.62 |
| | Our SM (iso-compute) | 0.5M | 1M | **0.316** | **0.667** | **0.559** | **0.766** | **29.90** | **18.08** | **11.40** | **17.29** |
| | *Gain* | | | *+0.241* | *+0.468* | *+0.267* | *+0.355* | *-12.63* | *-12.97* | *-10.35* | *-11.33* |
| AR | | | | | | | 0.760 | | | | 12.1 |
| Data | | | | | | | 1.0 | | | | 14.8 |


The same holds for validation perplexity:
| Model | Gradient updates | Forward passes | Training tokens | OWT PPL |
| :--- | :--- | :--- | :--- | :--- |
| AR | 0.5M | 0.5M | 262B | 17.54 |
| SEDD (Lou et al., 2024)  | 1M | 1M | 262B | $\leq24.10$ |
| MDLM (Sahoo et al., 2025) | 1M | 1M | 262B | $\leq23.21$ |
| GIDD+ (Rütte et al. 2025)| 1M | 1M | 262B | $\leq22.29$ |
| Our MDLM+SM (iso-compute) | 0.5M | 1M | 131B | $\leq22.36$ |
| Our MDLM+SM (iso-update) | 1M | 2M | 262B | $\leq\mathbf{21.47}$ |

We are still waiting for the unconstrained generation results for the iso-update model (1M steps), and will update the paper again as soon as the evaluation is completed.

---

# Updated results on MBPP+
Independent of the reviews, we identified a bug in the external lm-evaluation-harness used for MBPP+: the framework was incorrectly running only a single test case per problem rather than the full suite. As a result, our initially reported results on MBPP+ were too high. We have resolved this issue and updated Table 3, where SM's gains remain as before. Moreover, we will initiate a pull request to the GitHub project in order to fix this bug.

---

> ### Author Response · Authors · 2025-11-26
> **Additional results for unconstrained generation with fully trained SM model (iso-update)**
>
> As promised in our previous response, we have now updated the paper to include the unconstrained generation results for the from-scratch pretrained model under the iso-update training budget (1M steps).
>
> Interestingly, the iso-compute SM model (with $N/2$=0.5M pretraining steps) even slightly outperforms the iso-update model at certain lower NFE budgets (e.g., NFE $=1/4$ for standard unmasking and NFE $\leq 1/2$ for ReMDM). Thus, we observe that SM is particularly effective in compute-restricted training regimes.
>
>
> | Unmasking | Feedback | Updates | Fwd passes | MAUVE 1/8 | MAUVE 1/4 | MAUVE 1/2 | MAUVE 1/1 | PPL 1/8 | PPL 1/4 | PPL 1/2 | PPL 1/1 |
> | :--- | :--- | :--- | :--- | :--- | :--- | :--- | :--- | :--- | :--- | :--- | :--- |
> | Standard | Binary | 1M | 1M | 0.017 | 0.025 | 0.036 | 0.034 | 60.02 | 54.95 | 52.36 | 50.46 |
> | | Our SM (iso-compute) | 0.5M | 1M | 0.143 | **0.417** | 0.498 | 0.596 | 41.08 | 31.97 | 27.36 | 24.63 |
> | | *Gain* | | | *+0.126* | *+0.392* | *+0.462* | *+0.562* | *-18.93* | *-22.98* | *-24.99* | *-25.83* |
> | | Our SM (iso-update) | 1M | 2M | **0.155** | 0.383 | **0.535** | **0.602** | **39.61** | **30.74** | **26.12** | **23.53** |
> | | *Gain* | | | *+0.138* | *+0.358* | *+0.499* | *+0.568* | *-20.41* | *-24.21* | *-26.23* | *-26.93* |
> | ReMDM | Binary | 1M | 1M | 0.075 | 0.199 | 0.292 | 0.411 | 42.53 | 31.05 | 21.75 | 28.62 |
> | | Our SM (iso-compute) | 0.5M | 1M | **0.316** | **0.667** | **0.559** | 0.766 | 29.90 | 18.08 | 11.40 | 17.29 |
> | | *Gain* | | | *+0.241* | *+0.468* | *+0.267* | *+0.355* | *-12.63* | *-12.97* | *-10.35* | *-11.33* |
> | | Our SM (iso-update) | 1M | 2M | 0.263 | 0.626 | 0.511 | **0.774** | **29.62** | **17.58** | **10.85** | **16.72** |
> | | *Gain* | | | *+0.189* | *+0.427* | *+0.219* | *+0.363* | *-12.91* | *-13.48* | *-10.90* | *-11.90* |
> | AR | | | | | | | 0.760 | | | | 12.1 |
> | Data | | | | | | | 1.0 | | | | 14.8 |
>
> With these final results now included, we believe we have fully addressed the points raised in the reviews. We kindly invite you to review our full rebuttal and the updated manuscript. We look forward to engaging in a discussion with you regarding these new findings and addressing any remaining questions.

---

### Author Response · Authors · 2025-12-02
**Summary of the Discussion Phase**

We want to thank again everyone involved in dedicating their valuable time to evaluate our work. We are motivated by the positive initial reviews, and summarize how we addressed all remaining questions in this response. This list orders related feedback from multiple reviewers to highlight the major enhancements to the manuscript.

**1. Established fairness via "iso-compute" training**

*Addressed questions from reviewers wkhr & GihD*

- **Question:** Reviewers noted that soft-masking (SM) requires two forward passes per step, making a training-compute matched comparison to baselines difficult.
- **Response:** We introduced an **iso-compute training** regime. We trained the SM model for half the number of steps to match the FLOPs of the baseline binary masking model.
- **Result:** Even with half the training steps, the iso-compute SM model consistently outperforms the fully trained baseline in both language modeling (MAUVE scores and PPL) and coding tasks. This proves SM's training efficiency.

**2. Demonstrated capability for pre-training from scratch**

*Addressed questions from reviewers E5oT & GihD*

- **Question:** The initial submission focused on efficient adaptation using finetuning or pretraining continuation. Reviewers questioned if SM was suitable for pretraining and requested evidence.
- **Response:** We added comprehensive experiments pretraining a model from scratch using SM.
- **Result:** The SM model trained from scratch achieves superior unconditional generation (MAUVE) and validation perplexity compared to standard MDLM and SEDD baselines. This confirms SM is a fundamental improvement to the diffusion framework, not just a finetuning technique.

**3. Clarified decoding dynamics: "guided exploration" vs. "bias"**

*Addressed questions from reviewers rBVW & E5oT*

- **Question:** Reviewers hypothesized that SM might work by simply "memorizing" previous predictions or "locking in" greedy choices, raising valid questions about generalization.
- **Response:** We added a trajectory analysis (Appendix C.5) visualizing token probabilities over time.
- **Result:** The analysis shows that the model frequently overrides high-confidence SM hints when the denoising process resolves better semantic information. This confirms SM acts as guided exploration rather than a rigid bias, allowing the model to self-correct during generation.

**4. Strengthened theoretical derivation**

*Addressed question from reviewers rBVW*

- **Question:** No formal derivation of the training objective was provided.
- **Response:** We updated Section 3.2 to explicitly derive how the two-pass method approximates the intractable marginal distribution to minimize the NELBO.

**5. Profiling**

*Addressed question from reviewer GihD*

- **Question:** What is the inference cost overhead?
- **Response:** We profiled the model (results added to Appendix C.4).
- **Result:** SM adds only a marginal ~16% overhead per forward pass, which is outweighed by the performance gains.

**6. Evaluation fixes**

*Addressed question from reviewer E5oT*

- **Question:** MAUVE scores of baseline and continued pretrained models were lower than reported in literature.
- **Response:** We identified and fixed a batched-sampling bug in the baseline codebase (aligning our results with literature). Additionally, we identified a test-case bug in the MBPP+ external evaluation harness. We will report these issues to the corresponding GitHub projects.
- **Result:** While absolute scores shifted to align with the results reported in literature, SM's relative gains remain robust.

**7. Comparison to related work**

*Addressed questions from reviewers rBVW & wkhr*

- **Question:** wkhr requested a clearer distinction from "self-conditioning" and rBVW requested references showing superiority of diffusion over AR.
- **Response:**
   - SM vs. self-conditioning: We clarified that unlike self-conditioning (which concatenates inputs in continuous space), SM operates in discrete space, preserves input dimensionality (superposition), and uses a unique confidence-based weighting mechanism.
   - Diffusion vs. AR: We added citations and analysis highlighting domains where DLMs demonstrate superiority over AR, specifically in accelerated generation (e.g., Mercury), controllable generation (infilling/constraints), and non-causal reasoning tasks (e.g., planning and reverse reasoning).

**8. Ablation on top-k selection**

*Addressed question from reviewer GihD*

- **Question:** Questions regarding the sensitivity and task-dependence of the k parameter.
- **Response:** We clarified that optimal $k$ is task-dependent: lower $k$ suits constrained tasks (e.g., coding), while higher $k$ aids exploration in open-ended generation. We also experimented with a learnable $k$ (differentiable softmax) in language modeling, which yielded minor PPL improvements but at the cost of higher memory/compute.

---

### Meta-Review · Area_Chair_xJjB · 2026-01-06

**Summary:**

The key decision factors were (i) whether soft-masking (SM) provides a principled and generalizable improvement over binary masking in masked diffusion language models, (ii) whether the reported empirical gains remain valid under strict iso-compute comparisons, from-scratch pretraining, and downstream fine-tuning scenarios, and (iii) whether SM introduces practical overheads or inductive biases that could undermine generalization. Across reviews, there was broad agreement that the paper addresses a real and well-motivated weakness of binary masking—information loss due to hard mask states—and proposes a simple, elegant remedy. The final decision was primarily driven by the strength and completeness of the rebuttal, which convincingly demonstrated that SM’s benefits are not an artifact of extra compute, particular training regimes, or narrowly chosen tasks, but rather reflect a robust and broadly applicable improvement to the MDLM framework.

**Reviewer Concerns:**

Addressed concerns:
The rebuttal comprehensively addressed the major technical and empirical questions raised by reviewers. Most importantly, the authors introduced a clear iso-compute training protocol, showing that SM continues to outperform binary masking even when matched for forward-pass FLOPs by halving the number of training steps. This resolved fairness concerns around the two-pass training procedure. The authors further demonstrated that SM is not merely an efficient fine-tuning trick, but can be used for pretraining from scratch, where it yields consistent gains in unconditional generation quality (MAUVE) and validation perplexity relative to standard MDLMs and strong baselines such as SEDD and ReMDM.

Concerns about SM acting as a form of greedy bias or memorization were addressed through detailed decoding-trajectory analyses, which showed that SM provides guided exploration rather than locking in early predictions: the model frequently overrides high-confidence SM hints as more semantic information becomes available. The training objective was clarified with an explicit derivation showing how the two-pass procedure approximates the intractable marginal and still optimizes a valid NELBO surrogate. Questions about overhead were resolved with concrete profiling, showing only a modest per-forward-pass cost (~16%), which is outweighed by the quality gains. Baseline inconsistencies in MAUVE and MBPP+ were traced to concrete evaluation bugs and corrected, aligning results with prior literature while preserving SM’s relative improvements.

Remaining concerns:
After the rebuttal, no substantive technical concerns remained. Some reviewers noted that the choice of the top-k parameter is task-dependent and that a fully learnable variant incurs additional cost, but this was framed as a practical trade-off rather than a flaw. Likewise, while SM is conceptually related to self-conditioning, the authors clearly articulated the distinctions in state space, dimensionality, and confidence-based weighting, and positioned SM as a discrete, dimension-preserving analogue rather than a rebranding of prior methods.

**Reviewer Scores:**

wkhr: likely 6
rBVW: likely 6
GihD: likely 8
E5oT: likely 6

---

### Decision · Program_Chairs · 2026-01-26

Accept (Poster)